# Circular single-stranded DNA as a programmable vector for gene regulation in cell-free protein expression systems

Zhijin Tian[1,2,7], Dandan Shao[3,7], Linlin Tang[2,3,7], Zhen Li[3], Qian Chen[4], Yongxiu Song[2,5], Tao Li [1], Friedrich C. Simmel [6] & Jie Song [2,3] ✉

Cell-free protein expression (CFE) systems have emerged as a critical platform for synthetic biology research. The vectors for protein expression in CFE systems mainly rely on double-stranded DNA and single-stranded RNA for transcription and translation processing. Here, we introduce a programmable vector - circular single-stranded DNA (CssDNA), which is shown to be processed by DNA and RNA polymerases for gene expression in a yeast-based CFE system. CssDNA is already widely employed in DNA nanotechnology due to its addressability and programmability. To apply above methods in the context of synthetic biology, CssDNA can not only be engineered for gene regulation via the different pathways of sense CssDNA and antisense CssDNA, but also be constructed into several gene regulatory logic gates in CFE systems. Our findings advance the understanding of how CssDNA can be utilized in gene expression and gene regulation, and thus enrich the synthetic biology toolbox.

Cell-free protein expression (CFE) systems have become an attractive alternative platform for engineering and designing biological systems in vitro[1–3], and have thus been developed for various applications, including high-throughput drug screening[4], point-of-care diagnostics[5–10], biomanufacturing, and artificial cellular communication[11–16]. Vectors for protein expression in CFE systems are commonly circular double-stranded DNA molecules such as plasmids, or linear double-stranded DNA, which is usually obtained as a polymerase chain reaction (PCR) product[17]. Those DNA double-strands are vectors for typical DNA-dependent RNA polymerases, and are usually exerted via transcription factors for the regulation of transcription[18,19]. Typically, manipulating the integrity of the T7 promoter to alter the interaction between T7 RNA polymerase (RNAP) and its promoter has been used to control gene expression in vitro[9,10,20–24]. Although switchable molecular devices have been achieved by tuning the recognition and binding of transcription factors on the double-

stranded DNA, the limited number of available regulatory factors translates to a relatively low number of methods for regulating double-stranded DNA expression. This makes it difficult to construct more complex genetic circuits based on double-stranded DNA vectors alone[18,19,25–28].

On the other hand, RNA-based CFE systems have increasingly received attention due to the abundant availability of RNA secondary structures and the possibility to manipulate these structures via hybridization and strand displacement reactions[6,29,30]. Riboswitches and toehold switches are the most representative types of RNA-based CFE systems at the post-transcriptional level that allow to activate or repress gene expression through structural transformations triggered by DNA or RNA sequences or other target molecules[31–33]. Recently RNA regulators have been successfully engineered to enable relatively complex control of gene expression reactions, which holds great promise for application in biosensing and molecular diagnostics[8,34–36].

[1]Department of Chemistry, University of Science & Technology of China, Hefei, Anhui 230026, China. [2]Hangzhou Institute of Medicine, Chinese Academy of Sciences, Hangzhou, Zhejiang 310022, China. [3]Institute of Nano Biomedicine and Engineering, Department of Instrument Science and Engineering, School of Electronic Information and Electrical Engineering, Shanghai Jiao Tong University, Shanghai 200240, China. [4]College of Forestry, Northeast Forestry University, Harbin 150040 Heilongjiang, China. [5]Ningbo institute of Dalian University of Technology, Ningbo 315016, China. [6]Department of Bioscience, School of Natural Sciences, Technische Universität München, Garching, Germany. [7]These authors contributed equally: Zhijin Tian, Dandan Shao, Linlin Tang. ✉e-mail: songjie@him.cas.cn

However, the fast degradation and resulting poor stability of RNA molecules in solution[37,38], currently limits their wider applications.

Herein, we introduce circular single-stranded DNA (CssDNA) as a programmable vector for gene expression in CFE systems and demonstrate that CssDNA can provide another route for gene regulation. CssDNA, derived from the M13 bacteriophage, is already widely used as a DNA scaffold for the formation of DNA nanostructures of various geometries due to its addressability and programmability[39,40]. The scaffold DNA is folded along predetermined paths with the aid of hundreds of short-staple strands[41-44]. Furthermore, it has recently been demonstrated that genes encoded on circular single-stranded scaffolds with a custom (non-M13) sequence and folded within DNA origami can be expressed in mammalian cells[45,46]. To expand the application of CssDNA to cell-free systems, we specially designed a CssDNA containing a T7 promoter sequence and a protein-coding sequence, which enables effective protein expression in a yeast extract-based CFE system. We construct sense and antisense CssDNA and compare the differences between their pathways for protein expression. To conduct a more detailed investigation into the regulation of CssDNA expression, we analyze the impact of expression components such as dNTPs and enzyme inhibitors, as well as the secondary structure, particularly in the promoter region. Using the identified expression pathways, we construct and implement various logic gates as examples to demonstrate the feasibility of using CssDNA as a programmable vector for gene regulation in CFE systems.

## Results and Discussion

### Gene expression of CssDNA in the CFE system

To evaluate the performance of the CssDNA vector in the CFE system, we designed two customized versions of CssDNA, CssDNA(+) and CssDNA(−), containing a T7 promoter region and an enhanced green fluorescence protein (EGFP) coding region. CssDNA(+) presented the expression cassette encoding the EGFP sense strand from 5' to 3' (Fig. 1a, left). In contrast, CssDNA(−) contained the complementary sequence of the expression cassette encoding EGFP antisense strand in the direction of 3' to 5' (Fig. 1a, right). We created these customized CssDNA vectors using a pScaf phagemid containing an M13 origin of replication (M13 ori) and a mutant M13 ori (Supplementary Fig. 2)[47]. To characterize the CssDNA, we compared it to its corresponding plasmid using agarose gel electrophoresis and atomic force microscopy (AFM). As shown in Fig. 1b, a single gel band in the CssDNA lane (1605 nt), which migrated faster than the plasmid, indicating the high purity of CssDNA production. The AFM images of CssDNA displayed a curled structure resembling a ball of wool, whereas the structure of plasmid appeared more stretched, due to the high flexibility of single-stranded DNA instead of double-helical DNA (Fig. 1c)[48].

In this study, we utilized a commercially available cell-free gene expression system based on yeast extract[49]. We tracked the process of CssDNA protein expression using a microplate reader and measured EGFP fluorescence intensity in real time. Our results show that the target protein can be effectively expressed in the CFE system from both CssDNA(+) and CssDNA(−) gene vectors (Fig. 1d, Supplementary Fig. 3). The production of EGFP from both vectors gradually increased with prolonged reaction time, reaching a plateau (Fig. 1e, f). CssDNA(+) exhibited a comparable EGFP fluorescence curve as CssDNA(-), but CssDNA(−) took less time to reach the plateau, suggesting that its expression rate was faster than for CssDNA(+). We extracted the maximum expression rate constants from the fluorescence curves to quantitatively compare the two CssDNA vectors (Supplementary Fig. 4). When the protein expression level of the CssDNA vector reached a plateau (i.e. after the expression had stopped), we also purified the produced protein and quantified its yield (Supplementary Fig. 5). In addition, we also compared the expression level of CssDNA template with that of traditional expression template (plasmid), and

optimized this cell-free expression system to improve the expression level of CssDNA, as shown in Supplementary Fig. 6.

### Effects of different expression components for CssDNA in CFE systems

To better understand the expression of CssDNA in the CFE system, we examined the influence of various expression components on CssDNA expression. Initially, we supplemented the original system with deoxynucleoside triphosphates (dNTPs) as substrates for DNA synthesis (Fig. 1g, upper panel). In living organisms, dNTPs are utilized by DNA polymerases as substrates for genome replication, and these polymerases are responsible for the replication and repair of cellular DNA[50,51]. After dNTPs were added, we observed a considerable increase in EGFP levels from both CssDNA(+) and CssDNA(−) vectors, which suggests that dNTPs effectively promote CssDNA gene expression in the CFE system (Fig. 1h, i, red, Supplementary Fig. 7). The speed of protein expression remained faster for CssDNA(−) than for CssDNA(+). Furthermore, we investigated the impact of aphidicolin, a tetracyclic diterpenoid, on gene expression in this system (Fig. 1g, bottom panel). Aphidicolin is a DNA polymerase inhibitor that thwarts cellular DNA synthesis by disrupting DNA polymerase activity[52]. The fluorescence signal decreased by over 50% after adding aphidicolin, indicating that aphidicolin effectively represses protein expression of both CssDNA vectors (Fig.1h, i, gray, Supplementary Fig. 7, 8). Furthermore, we also explored the effect of aphidicolin on the plasmid vector, and discovered that aphidicolin had no effect on the plasmid, compared to both CssDNA vectors (Supplementary Fig. 8). Based on these findings, we suggest that the expression of CssDNA is linked to DNA synthesis by DNA polymerases. Different expression components may impact the level of CssDNA protein expression, which establishes the foundation for the regulation of CssDNA gene expression.

### Effects of T7 promoter region for CssDNA in CFE systems

The T7 promoter region is a specific DNA sequence that is recognized by T7 RNAP and initiates transcription. The integrity of this promoter domain is a prerequisite for RNA transcription[53]. To explore the role of the T7 promoter sequence in CssDNA vector during gene expression in the CFE system, we added T7 complementary stands to the CssDNA(+) and CssDNA(−) vectors, respectively (Fig. 2a). Figure 2b, c demonstrated that the addition of T7 complementary strands corresponding to CssDNA(+) had no effect on CssDNA(+) expression, whereas the addition of T7 complementary strands to CssDNA(-) significantly improved CssDNA(−) protein expression (Supplementary Fig. 9, 10, 15). The gene expression level of CssDNA(-) responded strongly to the presence of T7 complementary strands, with the final EGFP fluorescence intensity increasing to three times the original. The promoting effect of the T7 complementary strands was also demonstrated by the maximum rate constants of the corresponding fluorescence curves (Supplementary Fig. 11). We also compared the effect of T7 complementary strand on the plasmid vector (Supplementary Fig. 12). As expected, the additional of T7 complementary strand didn't have any effect on plasmid expression. In addition, we also observed the level of protein expression when both T7 complementary strands and aphidicolin were presented (Supplementary Fig. 10). For the CssDNA(+) vector, gene expression was significantly inhibited under these conditions, similar to the effect of aphidicolin acting alone. Conversely, the inhibitory effect of aphidicolin on CssDNA(−) almost disappeared when T7 complementary strands were presented. In other words, the promoting effect of T7 complementary strands on CssDNA(-) was not affected by aphidicolin.

To investigate the function of T7 complementary strands of different lengths on CssDNA(−) gene expression and to determine the length of accessible T7 promoter, six variants of different lengths were designed to complement the CssDNA(−) promoter domain of 9 nt, 13 nt, 17 nt, 19 nt, 23 nt, 27 nt respectively (Fig. 2d). The T7 promoter

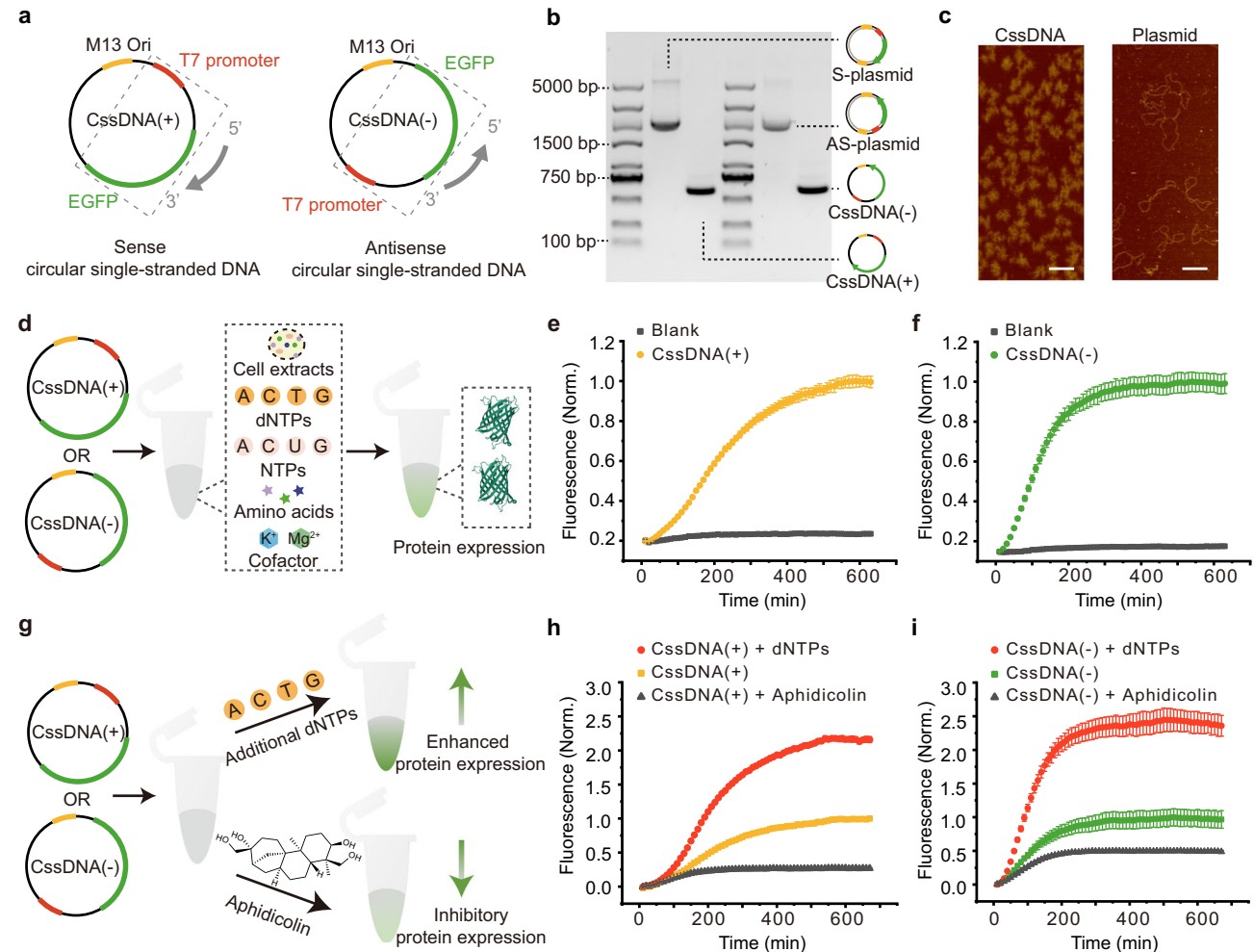

**Fig. 1 | Characterization of CssDNA and regulation of its protein expression in the CFE system. a** Design schematics of CssDNA vectors, where colors depict positions of vector features such as T7 promoter (red), EGFP (green) and M13 ori (orange). In sense circular single-stranded DNA (CssDNA(+), left), the coding sequence of the expression cassette is presented from 5' to 3'. In antisense circular single-stranded DNA (CssDNA(−), right), the template sequence of the expression cassette is in the reverse direction. **b** 1% agarose gel analysis of CssDNA(+) and its corresponding plasmid (S-plasmid), CssDNA(−) and its corresponding plasmid (AS-plasmid). **c** AFM images of CssDNA (left) and plasmid (right), scalebar 200 μm. **d** Schematic of gene expression of CssDNA(+) or CssDNA(−) vector in the CFE system. The CFE system contains all the essential components for CssDNA gene expression, with some representative components (not all) shown in the dashed box. **e, f** Changes in fluorescence signal over time for CssDNA(+) (**e**) and CssDNA(−) (**f**) vectors (5 ng/μL) during expression relative to the blank group. **g** Schematic representation of the regulation of CssDNA(+) or CssDNA(−) vector gene expression by dNTPs and aphidicolin in the CFE system. **h, i** Changes in protein expression levels over time for CssDNA(+) (**h**) and CssDNA(−) (**i**) vectors (5 ng/μL) in the presence of dNTPs or aphidicolin. All fluorescence signals were normalized according to the fluorescence intensity of the highest expression level of the corresponding expression vector. The maximum rate constant of the fluorescence kinetic curves in **e, f, h** and **i** was obtained by taking the first derivative of the corresponding curve, as shown in Supplementary Fig. 4. Data collected in **e, f, h** and **i** were monitored by a microplate reader and are presented as mean ± standard deviation (s.d.) for $n = 3$ biologically independent experiments, source data provided.

sequence used in this study was derived from the commercial optimized plasmid template pD2P, which contains 27 base pairs of the T7 promoter. The fluorescence curves of EGFP showed that T7 complementary strands of different lengths had distinct promoting effects on CssDNA(−) expression, in which the fluorescence enhancement rates were consistent (Fig. 2e, Supplementary Fig. 13). An in vitro RNA transcription kit was then used to further confirm how T7 complementary strands enhance CssDNA(−) gene expression. The gel showed that CssDNA(−) bound to its T7 complementary strands of different lengths can transcribe mRNA, but the transcription capacity of CssDNA(−) and CssDNA(+) themselves was negligible (Fig. 2f). To further quantify the amount of transcribed RNA, we determined the gray value intensity of the gel electrophoresis bands and also measured the RNA concentration via UV absorbance (Supplementary Fig. 14). The results show that the transcription capacity of CssDNA(−) + T7 (CssDNA(−) bound to T7 complementary strand)

decreased with decreasing T7 complementary strand length. The similar yield and rate of protein expression are due to the presence of excess templates in the reaction system, that is, the amount of RNA transcribed by CssDNA(−) + T7 is much greater than the RNA required for translation. This implies that the promoting effect of the T7 complementary strands can be attributed to transcription initiation and that a partially double-stranded CssDNA(−), formed by binding to its T7 complementary strands, can directly trigger transcription, even if it contains an accessible T7 promoter of only 9 bp[54–58]. The inhibition of CssDNA(−) expression by aphidicolin in the presence of different T7 complementary strands was negligible compared to CssDNA(−), suggesting that the RNA transcription process was not interfered with by aphidicolin, similar to the above results (Fig. 2g, Supplementary Fig. 16).

To investigate the impact of DNA strands complementing other domains outside the CssDNA(−) T7 promoter on gene expression,

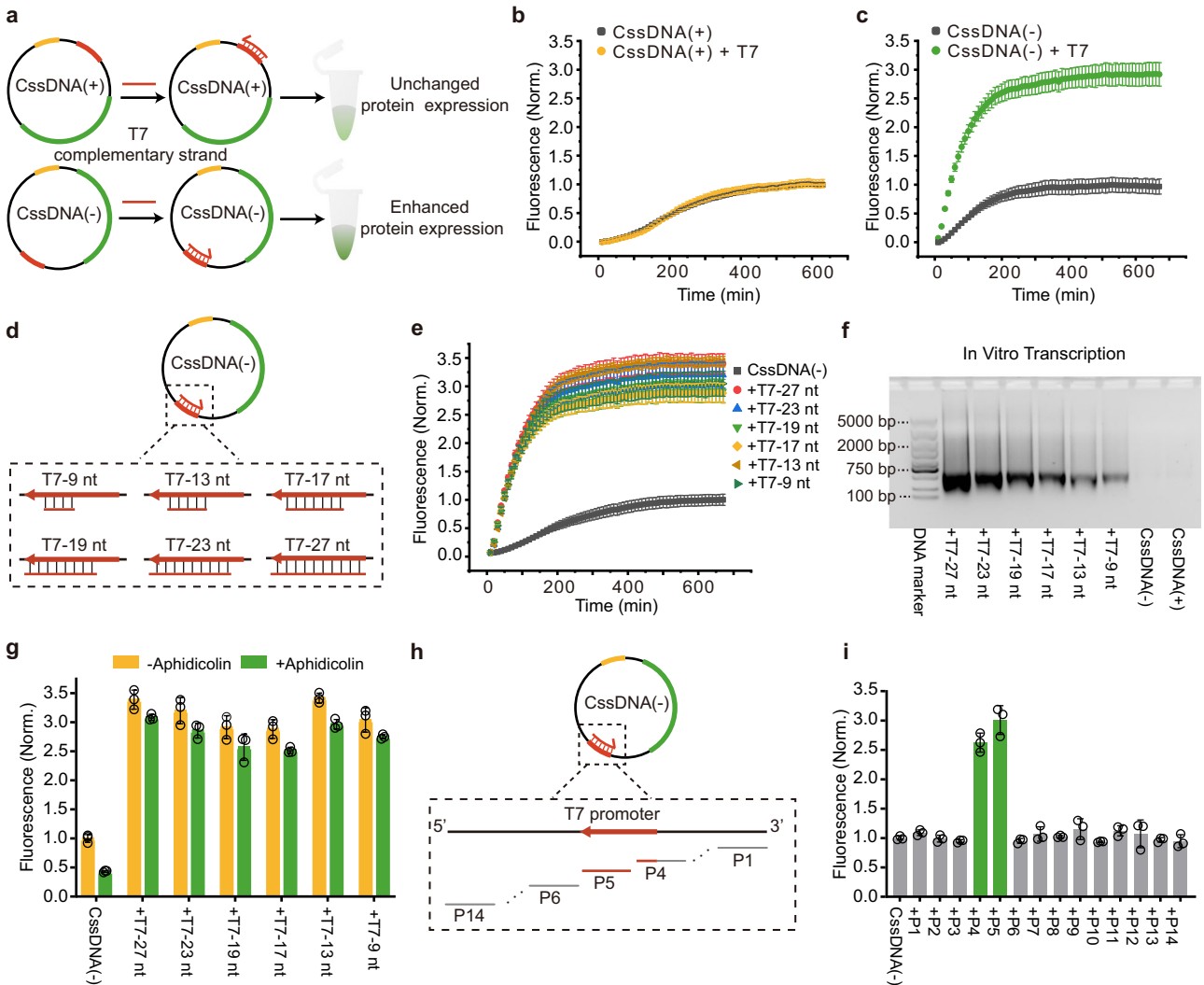

**Fig. 2 | The role of the T7 promoter region in CssDNA gene expression.**
**a** Schematic diagram illustrating how DNA strands complementary to corresponding CssDNA T7 promoter (T7 complementary strands) influence CssDNA gene expression levels. **b, c** Fluorescence signal changes over time for CssDNA(+) + T7 and CssDNA(-) + T7 (CssDNA bound to its T7 complementary strands), respectively, as compared to CssDNA(+) and CssDNA(-) alone. **d** Illustration of the binding of CssDNA(−) to T7 complementary strands of varying lengths, with the red section representing the 27 bp T7 promoter on CssDNA(·). **e** The changes in CssDNA(−) protein expression over time after the addition of T7 complementary strands of different lengths. The maximum rate constant of the fluorescence kinetic curves in **b, c** and **e** was obtained by taking the first derivative of the corresponding curve, as shown in Supplementary Fig. 11. **f** 1.5% agarose gel analysis of mRNA obtained by in vitro transcription of CssDNA(−) that combined with different T7 complementary strands, as well as CssDNA(+). **g** Fluorescence intensity of CssDNA(·) bound to T7 complementary strands during the protein expression plateau in the absence and presence of aphidicolin compared to CssDNA(−). **h** Schematic representation of the positions of the complementary strands on CssDNA(−), with the P5 region containing 19 bp T7 promoter sequence and the P4 region containing 8 bp sequence of the 27 bp T7 promoter front end. **i** Fluorescence intensity of CssDNA treated with or without other complementary strands during the expression plateau is shown. The CssDNA vector used here was 5 ng/μL. All fluorescence signals were normalized based on the fluorescence intensity of the corresponding CssDNA expression plateau. Data collected in **b, c, e, g** and **i** were monitored by a microplate reader and are presented as mean ± standard deviation (s.d.) for n = 3 biologically independent experiments, individual data points in **g** and **i** are overlaid, source data provided.

we designed another thirteen 19-nt DNA strands (denoted by P1, P2, P3...P14, respectively) around the T7 promoter region to hybridized with CssDNA(·) (Fig. 2h). Among them, P1-P4 were located in front of T7 promoter region, and P6-P14 were located behind T7 promoter. The results showed when regions P4 or P5, containing part of the T7 promoter sequence, were presented in the form of double-stranded DNA, gene expression was enhanced, while the other complementary strands had no impact on protein expression (Fig. 2i, Supplementary Fig. 17). Therefore, adding DNA strands that complement the T7 promoter in the CFE system is an efficient way to regulate CssDNA(−) gene expression by enabling CssDNA(·) to directly initiate transcription and enhance gene expression.

## Gene expression pathways of the CssDNA vector in the CFE system

Based on the results of Fig. 1 and Fig. 2, it is speculated that both the CssDNA(+) and CssDNA(−) vector expression processes are related to DNA synthesis, while the CssDNA(−) vector expression pathway differs from that of CssDNA(+), as it may also involve directly transcriptional mechanisms. To test this hypothesis, we conducted experiments using a series of concentrations of CssDNA vectors ranging from 0.5 ng/μL to 15 ng/μL, and monitored the corresponding EGFP expression reactions. Our system included the use of aphidicolin to inhibit DNA replication and the addition of T7 complementary strands to simulate the partially hybridized intermediates of DNA replication. The

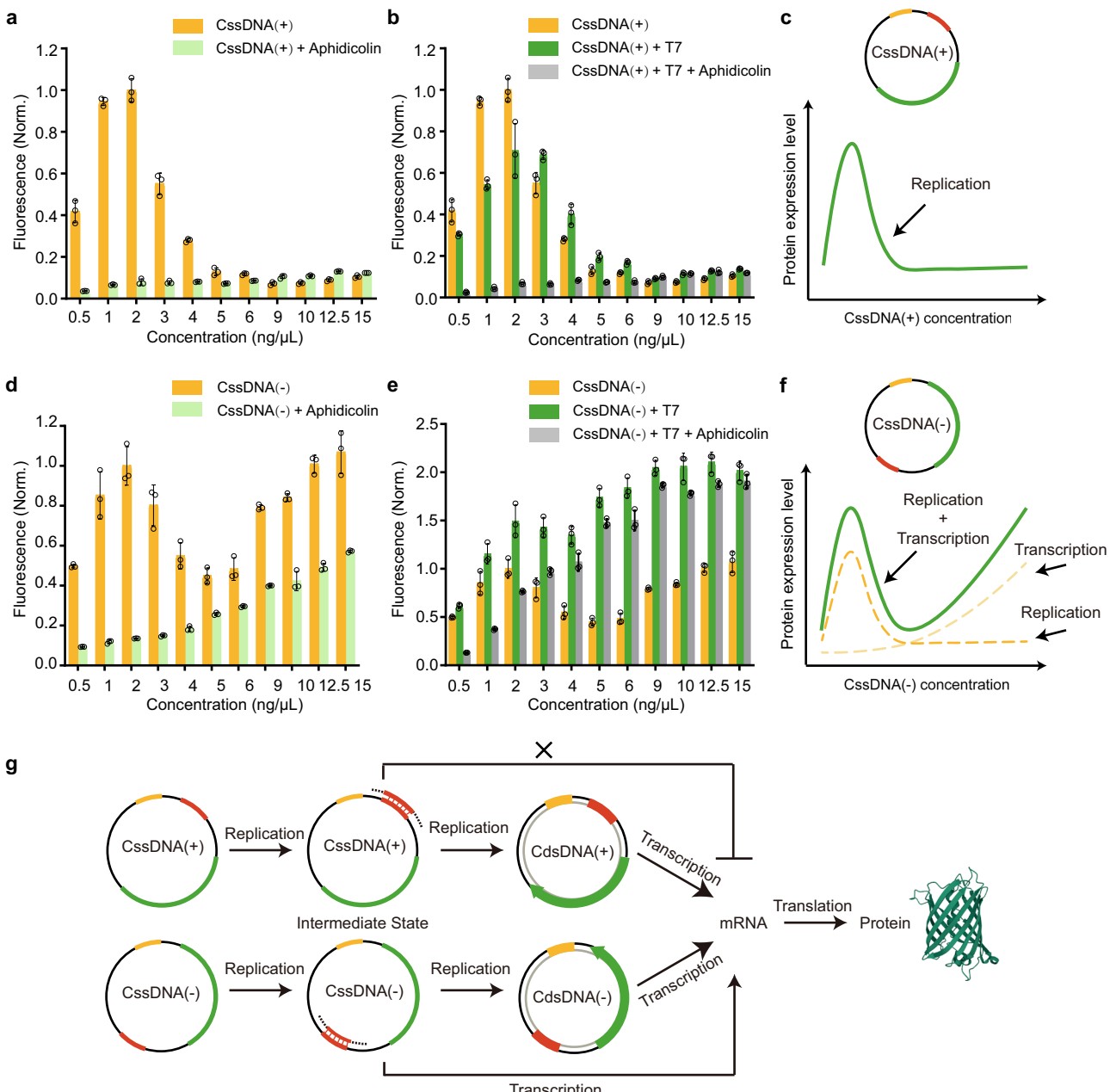

**Fig. 3 | Gene expression processes of two types of CssDNA vectors in the CFE system. a, d** EGFP fluorescence intensity of different concentrations of CssDNA(+) or CssDNA(−) vector produced at the protein expression plateau, compared to the corresponding CssDNA in the presence of aphidicolin. **b, e** Fluorescence intensity of EGFP produced by different concentrations of CssDNA(+) or CssDNA(−) when only T7 complementary strands are present or when T7 complementary strands coexist with aphidicolin. All fluorescence signals in **a** and **b**, **d** and **e** were normalized based on the average fluorescence of the corresponding CssDNA

expression plateau at a concentration of 2 ng/μL. Data collected in **a, b, d** and **e** were monitored by a microplate reader and are presented as mean ± standard deviation (s.d.) for $n = 3$ biologically independent experiments, individual data points are overlaid, source data provided. **c, f** The trend of protein expression level changes with the concentration of CssDNA(+) or CssDNA(−) vectors. **g** Schematic representation of the different protein expression pathways of CssDNA(+) and CssDNA(−) vectors.

methods we applied to simulate the intermediate processes of CssDNA expression made the expression pathway more explicit.

In the case of CssDNA(+) vector, EGFP production initially increased, and then decreased, as the vector concentration increased while all other reaction components were fixed. The maximum EGFP yield occurred at 2 ng/μL vector concentration (Fig. 3a, orange, Supplementary Fig. 18). Protein expression decreased at CssDNA(+) vector concentrations above 2 ng/μL, potentially due to a resource-sharing effect that results from a lack of substrates for DNA synthesis to generate complete double-stranded DNAs. The addition of aphidicolin to

the system disrupted DNA polymerase activity, interfering with the conversion of single-stranded DNA into double-stranded DNA and thus, resulting in significantly reduced protein production regardless of the CssDNA(+) vector concentration (Fig. 3a, light green, Supplementary Fig. 19). This showed that expression from incomplete circular double-stranded DNA (i.e., when the template strand is incomplete) produced by CssDNA(+) replication was negligible. Unsurprisingly, the addition of T7 complementary strands to CssDNA(+) vector at any concentration did not significantly enhance CssDNA(+) gene expression (Fig. 3b, dark green, Supplementary

Fig. 20). Consistent with prior results, EGFP yields in the presence of both T7 complementary strands and aphidicolin were as low as those in the presence of aphidicolin alone (Fig. 3b, gray, Supplementary Fig. 20). A protein expression curve of CssDNA(+) at various concentrations is displayed in Fig. 3c, indicating that complete double-stranded DNA synthesis via DNA replication is necessary for CssDNA(+) vector gene expression, followed by mRNA transcription and protein translation. In other words, protein expression from CssDNA(+) requires the synthesis of the complete complementary strand to act as a transcription template. This appears to be the sole pathway of CssDNA(+) gene expression (Fig. 3g, upper panel).

Interestingly, we observed a bimodal distribution of protein expression efficiency as a function of DNA concentration for the CssDNA(−) vector. Based on Fig. 3d, within the low concentration range from 0.5 ng/μL to 5 ng/μL, the optimal CssDNA(−) vector concentration for EGFP production was 2 ng/μL. EGFP production increased below 2 ng/μL and decreased above 2 ng/μL, similarly to the performance of the CssDNA(+) vector. By contrast, the gene expression level of the CssDNA(−) vector increased with increasing vector concentration at higher concentrations above 5 ng/μL (Supplementary Fig. 21). Another distinction between CssDNA(−) and CssDNA(+) was the effect of aphidicolin on the expression level at different concentrations of CssDNA(−) vector. Although aphidicolin significantly inhibited CssDNA(−) gene expression, EGFP levels tended to increase with increasing vector concentration after the addition of aphidicolin, rather than remaining at similarly low levels as for CssDNA(+) (Fig. 3d, light green, Supplementary Fig. 22). In sum, when the concentration of CssDNA(−) was higher, the expression of CssDNA(−) was increased, and the inhibitory effect of aphidicolin was weak. We have demonstrated that the inhibitory effect of aphidicolin acts on the replication process, but not on transcription (Figs. 1 and 2). Therefore, we conclude that not all expression of CssDNA(−) requires a complete replication process and that a transcription process is present. We thus surmise that, CssDNA(−) expression in the presence of aphidicolin is primarily due to direct transcription from the CssDNA starting from incomplete DNA replication intermediates (i.e., partially hybridized intermediates of DNA replication, in which the T7 promoter is present in double-stranded form), which is a gene expression pathway that differs from that for CssDNA(+). This can also be confirmed further by the analogous expression trend of CssDNA(−) at different concentrations to that of CssDNA(−) bound to T7 complementary strands, in the presence of aphidicolin (Fig. 3d, e, light green and gray). Similarly, the fluorescence intensity of EGFP monitored after the addition of T7 complementary strands was stronger than that of the corresponding CssDNA(−) itself, indicating that RNA transcription was activated (Fig. 3e, dark green, Supplementary Fig. 23). We observed that at low concentrations (especially below 2 ng/μL), the promotion of CssDNA(−) expression levels by the T7 complementary strand was weak, whereas at high concentrations, it was stronger (Fig. 3e, orange and dark green, Supplementary Fig. 23). It can also be observed that aphidicolin had a slight effect on gene expression of CssDNA(−) hybridized to T7 complementary strands (CssDNA(−) + T7) when CssDNA(−) was at high concentrations, which is consistent with our previous results (Fig. 2g). In contrast, the inhibitory effect was evident when CssDNA(−) + T7 was present at low concentrations (particularly below 2 ng/μL), which is similar to the effect on CssDNA(−) (Fig. 3e, dark green and gray, Supplementary Fig. 23). The differential promotion of CssDNA(·) by T7 complementary strand and the differential inhibition of CssDNA(·) + T7 by aphidicolin at different concentrations of CssDNA(−) vector underline that there are two expression pathways for CssDNA(−). Depending on the vector concentrations, these two pathways contribute differently to the overall expression level when the reaction components are given (Fig. 3f). The dominant pathway at low vector concentration appears to be complete DNA replication, whereas at high vector concentration, the dominant pathway is

transcription after incomplete replication (Fig. 3g, bottom panel). These results confirmed our hypothesis about the expression fate of the two CssDNA vectors in the CFE system and deepened our understanding of CssDNA as a programmable vector for the CFE system.

## Construction of logic gates in CFE systems using CssDNA

Having characterized the performance and gene expression pathways of two CssDNA vectors in the CFE system, we can now use them as components for the implementation of logic gates for gene regulation and biological computing. Due to the fast and strong response of CssDNA(−) to regulatory factors, particularly T7 complementary strands, we focused on CssDNA(·) as a logic element for logic gates (Fig. 4a). To achieve this, a set of ssDNAs were designed as inputs, and the fluorescence of EGFP was used as an output signal. Informed by our experiments on the expression pathways of CssDNA(−), we hypothesized that the addition of aphidicolin would inhibit replication-mediated gene expression of CssDNA(−), thereby improving the signal-to-noise ratio of logic gates (Fig. 4b).

In Fig. 2, we screened a variety of DNA strands complementary to the CssDNA(−) vector, and found several options for the design of inputs for an OR gate. From these variants, we selected complementary strands of 17 nt and 23 nt as input 1 and input 2, respectively. In the absence of either input, the CssDNA(−) vector produced EGFP at a low yield, reflecting an initial state of very low fluorescence. When one or both inputs were present, the fluorescence was at a high level due to the increased gene expression (Fig. 4c, Supplementary Fig. 24). As anticipated, the addition of aphidicolin to the reaction system significantly improved the signal-to-noise ratio, doubling the ratio between the 1 and 0 levels. (Fig. 4c, green).

To construct an INHIBIT (INH) gate, two alternative inputs were designed. Input 1, which contains the T7 complementary sequence that hybridizes to CssDNA(−) and a 3' flanking toehold sequence, can enhance gene expression. Input 2, which contains the fully complementary sequence to input 1, can displace input 1 from CssDNA(−) via a toehold-mediated strand displacement reaction. Thus, the concurrent presence of both inputs is expected to diminish fluorescence. As displayed in Fig. 4d, high fluorescence was only evident when input 1 was present alone; otherwise, fluorescence was low, which demonstrated that the INH logic gate was successfully implemented. In the presence of aphidicolin, the fluorescence intensity of output 1 was up to 10 times higher than that of output 0 (Fig. 4d, green, Supplementary Fig. 25). To design a NOR gate, we altered the initial gate structure by annealing the CssDNA(−) with a longer T7 complementary strand that had toehold sequences at both ends (Supplementary Fig. 26). Either input could release the CssDNA(·) from the gate structure via toehold at the 5' and 3' ends, respectively. Accordingly, fluorescence was high only in the absence of any input, but low in the presence of one or both inputs (Fig. 4e, Supplementary Fig. 27). The signal-to-ratio was also improved upon the addition of aphidicolin (Fig. 4e, green). In addition, we also constructed NAND and AND gates to demonstrate the general applicability of our approach (Supplementary Fig. 28). The implementation of logic gates further confirmed that CssDNA can serve as a programmable vector for gene regulation in a cell-free system.

In summary, we investigated gene expression using circular single-stranded DNA (CssDNA) as a programmable type of gene vector in CFE systems. We demonstrated that the expression level of CssDNA can be promoted or suppressed by additional components, such as dNTPs and aphidicolin, as well as by complementary strands of varying lengths and sequences. This highlights the regulatory potential of CssDNA in these systems. By varying the concentrations of CssDNA and simulating intermediate processes, we identified the differing expression fates of CssDNA(+) and CssDNA(−). CssDNA(+) follows a single expression pathway, in which fully complementary double-stranded DNA is synthesized through complete DNA replication followed by transcription. On the other hand, CssDNA(·) has two

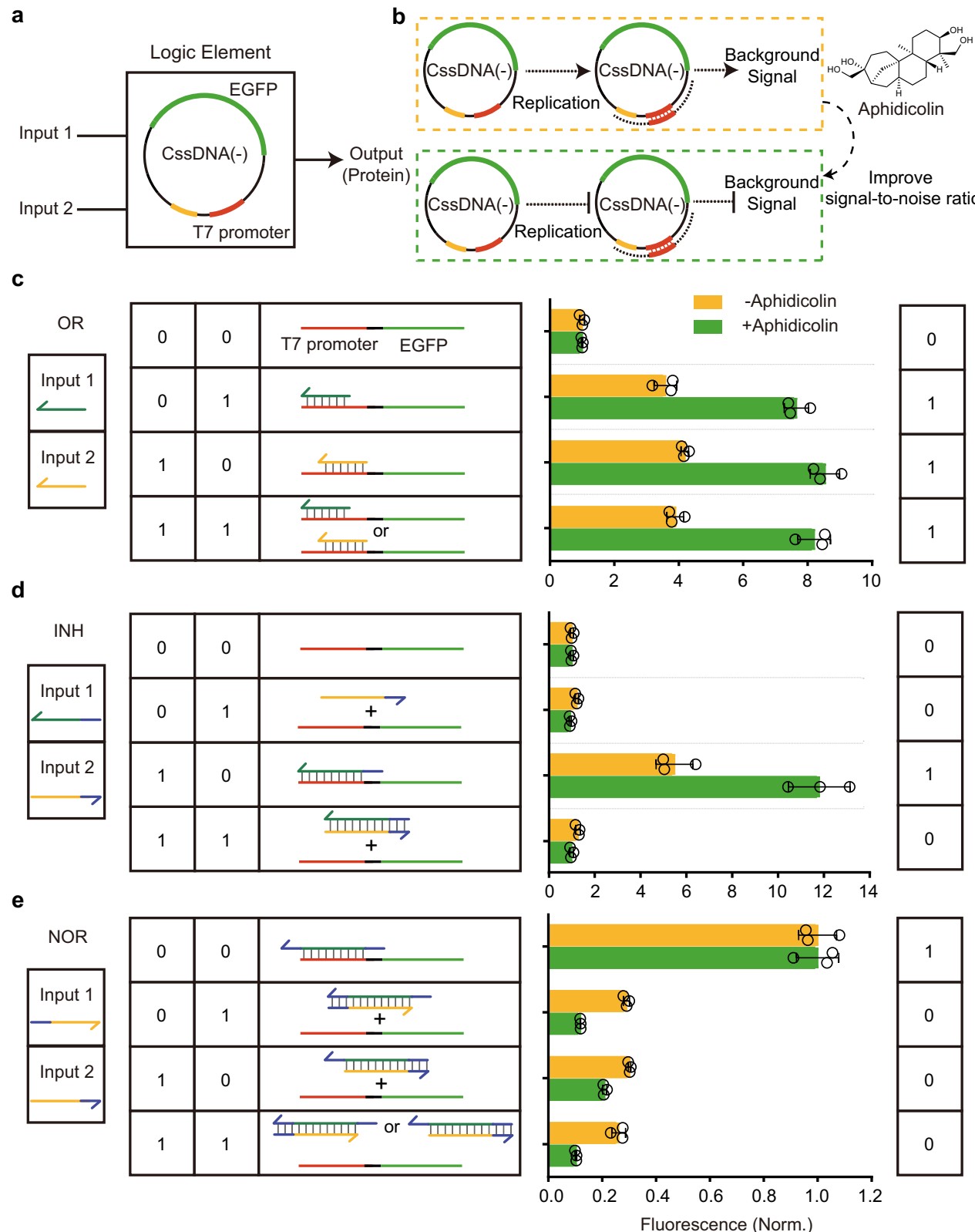

**Fig. 4 | Construction of logic gates using CssDNA as a logic element. a** A two-input logic gate was constructed using CssDNA(−) as the logic element and protein as the output. **b** Schematic illustration of the addition of aphidicolin to reduce the background signal from CssDNA(-) self-expression and to improve the signal-to-noise ratio of the logic gates. **c-e** Schematic and fluorescence signals of two-input logic gates under different input combinations, including OR, INHIBIT (INH) and

NOR. All fluorescence signals in **c, d** and **e** were normalized according to the fluorescence intensity of the corresponding initial gate structure expression plateau under no input conditions. Data collected in **c, d** and **e** were monitored by a microplate reader and are presented as mean ± standard deviation (s.d.) for $n = 3$ biologically independent experiments, individual data points are overlaid, source data provided.

expression pathways simultaneously, namely, complete DNA replication and incomplete DNA replication. The dominant pathway of CssDNA(−) primarily depends on the vector concentration when the reaction components are constant.

As an application example, two-input logic gates were designed and implemented using CssDNA(·) as the logic element. Interference with the DNA replication pathway resulted in an improved signal-to-noise ratio of the logic gates. Apart from logic gates construction, the CssDNA-based regulatory system holds great potential for biosensing and molecular diagnostics. As a gene expression vector, CssDNA enables programmable regulation of gene expression. This includes the formation of secondary structures through the use of staple strands, which can influence the different expression pathways of CssDNA[59]. Such a capability uniquely expands the toolbox of gene circuits and synthetic biology by the rich repertoire of methods previously employed in DNA nanotechnology.

## Methods

### Plasmid construction

The coding and template sequences of the EGFP expression cassette were amplified from plasmid pD2P (Kangma-Healthcode, Shanghai) using PCR. Primer sequences for PCR amplification can be found in Supplementary Table 1. The plasmid vector was obtained from pScaf-7560.1 (Addgene plasmid #111406) by digestion with two restriction endonucleases (KpnI and BamHI), which contains a M13 origin for ssDNA initiation and a modified M13 origin for the synthesis termination. All fragments were analyzed by agarose gel electrophoresis and purified using a gel extraction purification kit (CWBIO). The gene fragments were then assembled with pScaf vector fragments through Gibson assembly (Hieff Clone® Plus One Step Cloning Kit). The recombinant plasmids were verified by gene sequencing in Tsingke Biotechnology Co., Ltd, and then transformed into chemically competent DH5α *E.coli* cells for plasmid amplification.

### Circular single-stranded DNA (CssDNA) production

To produce custom CssDNA, chemically competent XL1-Blue *E.coli* cells were co-transformed with the recombinant plasmid and a helper plasmid (PSB4423). The resulting single colonies were grown for 18–20 h in 800 mL 2xYT media (1% tryptone, 0.5% yeast extract, 1% NaCl) containing 10 μg/mL ampicillin, 10 μg/mL chloramphenicol and 500 mM MgCl$_2$ (30 °C, 220 rpm). The culture was centrifuged at 8000 g for 20 min at 4 °C to collect the bacteria pellet, and the supernatant was harvested. Phage still present in the supernatant was precipitated by adding PEG 8000 and NaCl and shaking for 2 h at room temperature. The phage pellet was then collected by centrifugation at 17000 g for 40 min at 4 °C. The phage pellet was resuspended in 5 mL 1x TE buffer and centrifuged at 4000 g for 10 min to remove any remaining bacteria. The phage-containing supernatant was lysed using QIAGENE kit (EndoFree® Plasmid Maxi Kit) to extract CssDNA. The CssDNA(+) and CssDNA(−) vectors were obtained by this phagemid production, and their sequences can be found in Supplementary Table 2.

### Agarose gel analysis

The purified CssDNAs and corresponding plasmids were analyzed by running 1% agarose gel electrophoresis in 1x TE buffer (10 mM Tris-HCl, 1 mM EDTA, PH 8.0) for 40 min at a voltage of 110 V. To characterize the CssDNA and its primer assemblies, 1% agarose gel in 1x TE buffer and 5 mM Mg$^{2+}$ was used, and gel electrophoresis was performed in TE-Mg$^{2+}$ buffer for 1 h at 60 V. For agarose gel electrophoresis to characterize mRNA, gels and running buffers were made with DEPC water to prevent mRNA degradation. Different mRNA samples of the same volume were run in 1% agarose gel at 150 V for 30 min. All gels were imaged using the Amersham ImageQuant 8000.

### AFM imaging

For AFM imaging, 2 μL DNA sample was deposited onto freshly cleaved mica and left for 3 min to allow absorption to the surface. Subsequently, the sample was added with 4 μL water and the mica was dried with nitrogen. Under ambient condition, an ultra-sharp silicon probe with a spring content of 0.35 N/m was used to capture AFM images in AC air topography mode on the Cypher VRS system (Oxford instruments). All the images were analyzed and processed with AR analysis software.

### Cell-free protein expression

One tube of ProteinFactory fast reaction mixture (Kangma-Healthcode, Shanghai) was dissolved in 10 mL of water. Each experimental system contained 40 μL of the reaction mixture in a 96-well plate, and the total volume of a single experiment is 60 μL. After adding the desired concentration of CssDNA vector and brief centrifugation, the fluorescence intensity of EGFP was monitored in real time using a multi-function measuring instrument (Tecan SparkControl). The reaction was continuously shaken at 150 rpm at 28 °C and measured every 10 min. The reaction system was supplemented with 20 μL of water without CssDNA vector as a blank group, and the background signals of the blank group were subtracted from the signals of all subsequent experimental groups. Fluorescence inverted microscopy (OLYMPUS CKX53) was used for imaging after protein expression was stopped.

### Protein purification and quantification

We used His-Monster Beads (purchased from Kangma-Healthcode, Shanghai) to purify the produced His-tagged protein. When the protein expression level of CssDNA vector reached a plateau (that is, the expression stopped), 1 mL of the reaction mixture was taken and centrifuged at 4000 rpm for 3 min at 4 °C, and the supernatant was collected. After centrifugation, the supernatant containing EGFP was added to the magnetic beads and mixed at 4 °C for 1 h. The beads bound to the target protein were washed three times with a washing solution (20 mM Tris-HCl, 500 mM NaCl, 20 mM Imidazole, pH 8.0), and then the protein was eluted with an elution solution (20 mM Tris-HCl, 500 mM NaCl, 250 mM Imidazole, pH 8.0). The protein was quantified by bicinchoninic acid (BCA) assay after overnight dialysis. The result showed that when the final concentration of CssDNA(+) was 1 ng/μL, the protein yield was 10 mg/L.

### Effects of addition components on protein expression

Prior to measurement, additives were incorporated into the reaction system. In the present system, the final concentration of dNTPs was 0.17 mM each and the final concentration of aphidicolin was 0.33 mM. The complementary strands and CssDNA vector were simultaneously added to the reaction system at a 5: 1 stoichiometric ratio. Sequences of the complementary strands can be found in Supplementary Table 3. To investigate protein expression under distinct conditions and concentrations, the CssDNA vector was co-added with the additional components in the reaction mixture, and the total reaction volume was maintained at 60 μL.

### In vitro RNA transcription

T7 High Yield RNA Transcription Kit was utilized for the in vitro transcription of mRNA based on CssDNA vector. As instructed, the CssDNA vector was added at a concentration of 1 μg, and the complementary strands were added at a ratio of 5: 1 to CssDNA. Following 2 h of incubation at 37 °C, DNaseI was incorporated in order to digest the DNA vector for 15 min. We subsequently used the lithium chloride purification method to eliminate protein and most free nucleotides. The resulting RNA was then dissolved in RNase-free water. To compare results, RNAs derived from different vectors were dissolved in the same volume of water, and then different RNA samples of the same

volume were introduced into each agarose gel lane. The concentration of transcribed RNA was measured by nanodrop (Thermo Fisher).

## Assembly of CssDNA and complementary strands

To assemble CssDNA and the corresponding complementary strands, both were simultaneously added to 1x TE buffer containing 5 mM Mg$^{2+}$ in a ratio of 1: 5. Following this, the DNA samples were heated to 85 °C for 5 min, from 85 °C to 37 °C at the rate of 1 °C/5 min and then 37 °C for 1 h.

## Construction of logic gates

For the OR gate, both inputs were introduced to the reaction system at a ratio of 5: 1 to CssDNA(-). In the case of the INH gate, input 2 was added at a 5: 1 ratio to CssDNA, and input 1 at a ratio of 20: 1. As for the NOR gate, CssDNA(−) was first assembled with its complementary strands as described previously, and subsequently the input strand was introduced in a 4: 1 ratio to the complementary strands. Input was incubated with the initial structure at 37 °C for 3 h. For the construction of a NAND gate, the steps are similar. CssDNA(-) was first bound to the complementary strands, and then the input strand was introduced at 4 times the molar number of the complementary strands. For the AND gate, T7 blocking strand was introduced in a 4: 1 ratio to block the T7 complementary strand. The double-stranded DNA was formed by heating to 85 °C for 5 min, from 85 °C to 37 °C at the rate of 1 °C/5 min and then 37 °C for 1 h. Then this double-stranded DNA was introduced in a 4: 1 ratio with CssDNA(·). Input was incubated with the initial structure at 37 °C for 3 h. The sequences of all the DNA strands used to construct the logic gates are given in Supplementary Table 4.

## Statistics and reproducibility

Statistical analysis was performed using OriginPro 2021 and GraphPad Prism v10.0 (GraphPad Software). The data is illustrated as the mean ± deviation, and the individual data points representing biological replicates are shown. For comparisons of two groups, two-tailed $t$ tests were used. Differences were considered significant at $*p \le 0.05$, $**p \le 0.01$, $***p \le 0.001$, $****p \le 0.0001$. Exact $p$ values are indicated in the Source Data file. All samples presented in agarose gels are representative of $n = 3$ independent agarose gel electrophoresis repeats. All fluorescence microscopy images of expressed EGFP are representative of sample images on $n = 3$ biologically independent repeats. No data were excluded from the analysis. The Investigators were not blinded to allocation during experiments and outcome assessment.

## Reporting summary

Further information on research design is available in the Nature Portfolio Reporting Summary linked to this article.

# Data availability

The authors declare that the data supporting the findings of this study are available within the paper and its supplementary information. Source data for each graph and uncropped gel images are provided as a Source Data file. Source data are provided with this paper.

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

## Acknowledgements

The authors are grateful for the financial support from the National Key Research and Development Program of China (Grant:No.2021YFF1200200 and 2023YFF1204500, to J.S.), the National Natural Science Foundation of China (Nos.22161132008, to J.S.), the Starry Night Science Fund of Zhejiang University Shanghai Institute for Advanced Study (SN-ZIU-SIAS-006, to J.S.), and the Natural Science Foundation of Zhejiang Province (LQ21C050001, to J.S.). J.S. also acknowledges the support from Youth Cross-disciplinary Team Project of the ChineseAcademy of Sciences. F.C.S. acknowledges support through the Deutsche For- schungsgemeinschaft (DFG SI 761/5-1, to F.C.S.).

## Author contributions

J.S. and Z.T. developed the concepts. Z.T. was responsible for all experiments, results and discussions, and for drafting and revising the manuscript. D.S. constructed the protein expression vectors. L.T. provided guidance on experimental planning. Z.T. and L.T. carried out data analysis and discussion. D.S., Z.T. and Q.C. studied and optimized the expression system. Z.L. assisted with in vitro transcription experiments. Q.C. assisted in optimizing the expression system. Y.S. and Z.L. characterized DNA samples. T.L. and F.C.S contributed to the writing of the manuscript. J.S. supervised the project and edited the manuscript. All the authors discussed the results and approved the final version of the manuscript.

## Competing interests

The authors declare no competing interests.
