## [Peer Review File · Nature Communications]

REVIEWER COMMENTS

Reviewer #1 (Remarks to the Author):

Summary:

In this manuscript, the authors describe a novel cell-free gene expression system using circular single-stranded DNA (C_{ss}DNA) as a vector. They investigate the difference between different C_{ss}DNA templates, namely (+) and (-) orientations. They find that both C_{ss}DNA (+) and (-) expression can be enhanced by dNTP supplementation and suppressed by the addition of aphidicolin, a DNA polymerase inhibitor. C_{ss}DNA (-) is uniquely enhanced by the supplementation of T7 complementary ssDNA of any length, which also increases activity significantly in the presence of aphidicolin. The authors hypothesize that C_{ss}DNA (+) vector expression is based on DNA synthesis to a double stranded vector, while C_{ss}DNA (-) is related to both replication and transcription of partial replication fragments. They investigate this hypothesis by titrating C_{ss}DNA concentration with T7 strands and aphidicolin. Finally, they demonstrate the application of this technology by forming OR, INH, and NOR gates using C_{ss}DNA(-) as the expression template and toehold T7 complementary strands as the dual input system.

Overall, the development of a novel expression template and its potential use in expanding the toolbox for cell-free gene circuits is useful for the field. While the authors do a nice job, I am struggling to understand the true potential impact on the field and have some major concerns that lessened my enthusiasm. There are key questions about the mechanisms purported by the authors and some points that require far more clarification than is given.

Major Criticisms:

- How does the expression level of C_{ss}DNA compare to traditional methods (linear expression templates, plasmids, etc). The authors show plasmid expression in S4 with and without aphidicolin, but it would be useful to show how the technology compares to existing state of the art from the same experiment. Most values in the work are normalized, but a more quantitative measure of C_{ss}DNA would be useful addition for benchmarking purposes. IN additiong mg/L of protein produced is important to describe.
- If similar, I just am not sure if people would switch to using this approach, which lessens my enthusiasm for having the work published in a high impact journal.
- The authors claim that C_{ss}DNA (+) is not able to be transcribed directly, but rather must be converted to dsDNA by DNA polymerase first, and then transcribed. It would be useful to prove this directly by in vitro transcription reaction, as was done for C_{ss}DNA (-). Also, if this is true, why not just use double stranded DNA?
- The discussion around the mechanism proposed in Fig 3g with incomplete DNA replication intermediates is not well explained. The term "DNA replication intermediates" is suddenly introduced

and is poorly defined. It seems that the authors are saying some of these replication intermediates will look like T7 complementary strands fused to CssDNA through chance, and that subset can be directly transcribed? The authors should explain this section more thoroughly and give more explicit reasonings behind their interpretation of the data in Fig 3.

Minor Criticisms

- In lines 49-42 the authors claim that it is “infeasible” to create complex genetic circuits from double stranded DNA and cite a paper on mammalian circuits and allosteric transcription factors. This is a large claim to make with minimal support. Please reword or expand with better citations.
- Do the authors have hypotheses on why CssDNA (+) and (-) behave so differently? Do the authors have an idea for the mechanistic reason that CssDNA (+) is unable to be transcribed directly (or from intermediates/T7 complementary strand fusion) and must only rely on replication? Why are the expression pathways different?
- There is no Y axis title for Fig S10.
- Further explain what the normalization scheme is for fluorescence. What the “equivalent Css DNA expression plateau” is is not clear.
- In line 177 the authors state there are 13 DNA strands designed, but list 14 designs P1-P14. Is this a typo or is one a control?

Reviewer #3 (Remarks to the Author):

Tian et al present a novel component for cell-free expression systems in their manuscript in the form of circular single stranded DNA (CssDNA) templates. They study the protein expression kinetics of CcssDNA templates carrying sense and antisense genes and identify the dominant pathways for both and different components, such as complementary nucleotides and DNA synthesis inhibitors that they can use to exploit these to create logic gates, demonstrating CcssDNA’s applicability in reaction circuits. In my opinion the CcssDNA templates presented in this work can be a useful, new addition in the toolbox for generating gene circuits that have clear advantages compared to double stranded templates as they can be more easily be integrated into oligonucleotide reaction circuits therefor, I find the works contribution

to the field of synthetic biology significant. I have found the experiments performed to be quite thorough and convincing and to support the conclusions drawn upon them. However, I have a number of comments and points the authors should address before I would recommend the manuscript to be accepted for publication.

1. The authors use phage derived ssDNA as their template. While it is quite an efficient way to produce ssDNA, in my opinion it potentially introduces some unneeded complexity. There are other ways to generate long ssDNA molecules that would only or mostly contain the gene-cassette e.g., with RCA (Nat Methods 10, 647–652 (2013)), asymmetric (Sci Rep 9, 6121 (2019)) or other PCR approaches (Nano Lett. 2009, 9, 12, 4302–4305) or by using a much smaller scaffold to clone in (Nanoscale, 2015,7, 16621-16624). If the template would only or mostly contain the gene some of the observed concentration dependent negative effects would be reduced possibly as more products would be complete. While the comparison with templates generated with the RCA and mini scaffold approach is out of the scope of this work, I think it would be relatively easy to produce a single stranded templates encompassing the gene cassette using one of the PCR approaches and compare its behavior to the CssDNA, therefore I think the authors should perform these experiment in addition to comparing their approach to these alternative approaches in the text.

2. The authors perform a number of experiments (e.g. in Fig.1 e/f/h/l, Fig.2 b/c/e) where they follow the gene expression in time and report on the differences in the observed kinetics. The authors should fit these curves to derive apparent kinetic parameters (e.g. rate constants) and report those in the text.

3. The results presented in Fig. 2e/f/g are somewhat perplexing. Based on the results presented in Fig2.f the 9nt and 13nt T7 oligos produce a much lower amount of RNA than the rest of the oligos. The other should by the way quantitate this and report it in a plot. Taken this difference in transcription yield it is quite remarkable the expression of the protein seems to have similar yield (based on the saturation value) and rate (based on the slope of the linear phase of the curve). The authors should comment on this.

4. The results presented in Fig. 4 are a nice demonstration of the system's capability to create logic gates. However, some fundamental logic gates, such as AND gates, are not demonstrated, which could be important to implement to show general applicability, using perhaps input oligos with partially complementary sequences to stabilize their binding to the promoter through three way junctions. Additionally, the authors claim that they create gene circuits using CssDNA when they create only logic gates in the strict sense. The authors should change this in the text.

REVIEWER COMMENTS

Reviewer #1 (Remarks to the Author):

Summary:

In this manuscript, the authors describe a novel cell-free gene expression system using circular single-stranded DNA (CssDNA) as a vector. They investigate the difference between different CssDNA templates, namely (+) and (-) orientations. They find that both CssDNA (+) and (-) expression can be enhanced by dNTP supplementation and suppressed by the addition of aphidicolin, a DNA polymerase inhibitor. CssDNA (-) is uniquely enhanced by the supplementation of T7 complementary ssDNA of any length, which also increases activity significantly in the presence of aphidicolin. The authors hypothesize that CssDNA (+) vector expression is based on DNA synthesis to a double stranded vector, while CssDNA (-) is related to both replication and transcription of partial replication fragments. They investigate this hypothesis by titrating CssDNA concentration with T7 strands and aphidicolin. Finally, they demonstrate the application of this technology by forming OR, INH, and NOR gates using CssDNA(-) as the expression template and toehold T7 complementary strands as the dual input system.

Overall, the development of a novel expression template and its potential use in expanding the toolbox for cell-free gene circuits is useful for the field. While the authors do a nice job, I am struggling to understand the true potential impact on the field and have some major concerns that lessened my enthusiasm. There are key questions about the mechanisms purported by the authors and some points that require far more clarification than is given.

Major Criticisms:

- How does the expression level of CssDNA compare to traditional methods (linear expression templates, plasmids, etc). The authors show plasmid expression in S4 with and without aphidicolin, but it would be useful to show how the technology compares to existing state of the art from the same experiment. Most values in the work are normalized, but a more quantitative measure of CssDNA would be useful addition for benchmarking purposes. IN additiong mg/L of protein produced is important to describe.

- If similar, I just am not sure if people would switch to using this approach, which lessens my enthusiasm for having the work published in a high impact journal.

- The authors claim that CssDNA (+) is not able to be transcribed directly, but rather must be converted to dsDNA by DNA polymerase first, and then transcribed. It would be useful to prove this directly by in vitro transcription reaction, as was done for CssDNA (-). Also, if

this is true, why not just use double stranded DNA?

- The discussion around the mechanism proposed in Fig 3g with incomplete DNA replication intermediates is not well explained. The term "DNA replication intermediates" is suddenly introduced and is poorly defined. It seems that the authors are saying some of these replication intermediates will look like T7 complementary strands fused to CssDNA through chance, and that subset can be directly transcribed? The authors should explain this section more thoroughly and give more explicit reasonings behind their interpretation of the data in Fig 3.

Minor Criticisms:

- In lines 49-42 the authors claim that it is "infeasible" to create complex genetic circuits from double stranded DNA and cite a paper on mammalian circuits and allosteric transcription factors. This is a large claim to make with minimal support. Please reword or expand with better citations.

- Do the authors have hypotheses on why CssDNA (+) and (-) behave so differently? Do the authors have an idea for the mechanistic reason that CssDNA (+) is unable to be transcribed directly (or from intermediates/T7 complementary strand fusion) and must only rely on replication? Why are the expression pathways different?

- There is no Y axis title for Fig S10.

- Further explain what the normalization scheme is for fluorescence. What the "equivalent Css DNA expression plateau" is is not clear.

- In line 177 the authors state there are 13 DNA strands designed, but list 14 designs P1-P14. Is this a typo or is one a control?

Reviewer #3 (Remarks to the Author):

Tian et al present a novel component for cell-free expression systems in their manuscript in the form of circular single stranded DNA (CssDNA) templates. They study the protein expression kinetics of CcssDNA templates carrying sense and antisense genes and identify

the dominant pathways for both and different components, such as complementary nucleotides and DNA synthesis inhibitors that they can use to exploit these to create logic gates, demonstrating C_{ss}DNA's applicability in reaction circuits. In my opinion the C_{ss}DNA templates presented in this work can be a useful, new addition in the toolbox for generating gene circuits that have clear advantages compared to double stranded templates as they can be more easily be integrated into oligonucleotide reaction circuits therefor, I find the works contribution to the field of synthetic biology significant. I have found the experiments performed to be quite thorough and convincing and to support the conclusions drawn upon them. However, I have a number of comments and points the authors should address before I would recommend the manuscript to be accepted for publication.

1. The authors use phage derived ssDNA as their template. While it is quite an efficient way to produce ssDNA, in my opinion it potentially introduces some unneeded complexity. There are other ways to generate long ssDNA molecules that would only or mostly contain the gene-cassette e.g., with RCA (Nat Methods 10, 647–652 (2013)), asymmetric (Sci Rep 9, 6121 (2019)) or other PCR approaches (Nano Lett. 2009, 9, 12, 4302–4305) or by using a much smaller scaffold to clone in (Nanoscale, 2015,7, 16621-16624). If the template would only or mostly contain the gene some of the observed concentration dependent negative effects would be reduced possibly as more products would be complete. While the comparison with templates generated with the RCA and mini scaffold approach is out of the scope of this work, I think it would be relatively easy to produce a single stranded templates encompassing the gene cassette using one of the PCR approaches and compare its behavior to the C_{ss}DNA, therefor I think the authors should perform these experiment in addition to comparing their approach to these alternative approaches in the text.

2.The authors perform a number of experiments (e.g. in Fig.1 e/f/h/l, Fig.2 b/c/e) where they follow the gene expression in time and report on the differences in the observed kinetics. The authors should fit these curves to derive apparent kinetic parameters (e.g. rate constants) and report those in the text.

3.The results presented in Fig. 2e/f/g are somewhat perplexing. Based on the results presented in Fig2.f the 9nt and 13nt T7 oligos produce a much lower amount of RNA than the rest of the oligos. The other should by the way quantitate this and report it in a plot. Taken this difference in transcription yield it is quite remarkable the expression of the protein seems to have similar yield (based on the saturation value) and rate (based on the slope of the linear phase of the curve). The authors should comment on this.

4.The results presented in Fig. 4 are a nice demonstration of the system's capability to create logic gates. However, some fundamental logic gates, such as AND gates, are not

demonstrated, which could be important to implement to show general applicability, using perhaps input oligos with partially complementary sequences to stabilize their binding to the promoter through three way junctions. Additionally, the authors claim that they create gene circuits using CssDNA when they create only logic gates in the strict sense. The authors should change this in the text.

REVIEWER COMMENTS

Reviewer #1 (Remarks to the Author):

Summary:

In this manuscript, the authors describe a novel cell-free gene expression system using circular single-stranded DNA (CssDNA) as a vector. They investigate the difference between different CssDNA templates, namely (+) and (-) orientations. They find that both CssDNA (+) and (-) expression can be enhanced by dNTP supplementation and suppressed by the addition of aphidicolin, a DNA polymerase inhibitor. CssDNA (-) is uniquely enhanced by the supplementation of T7 complementary ssDNA of any length, which also increases activity significantly in the presence of aphidicolin. The authors hypothesize that CssDNA (+) vector expression is based on DNA synthesis to a double stranded vector, while CssDNA (-) is related to both replication and transcription of partial replication fragments. They investigate this hypothesis by titrating CssDNA concentration with T7 strands and aphidicolin. Finally, they demonstrate the application of this technology by forming OR, INH, and NOR gates using CssDNA(-) as the expression template and toehold T7 complementary strands as the dual input system.

Overall, the development of a novel expression template and its potential use in expanding the toolbox for cell-free gene circuits is useful for the field. While the authors do a nice job, I am struggling to understand the true potential impact on the field and have some major concerns that lessened my enthusiasm. There are key questions about the mechanisms purported by the authors and some points that require far more clarification than is given.

We thank reviewer #1 for their constructive remarks on our work. Based on these comments, we have made the following changes and further improved the manuscript.

Major Criticisms:

- How does the expression level of CssDNA compare to traditional methods (linear expression templates, plasmids, etc). The authors show plasmid expression in S4 with and without aphidicolin, but it would be useful to show how the technology compares to existing state of the art from the same experiment. Most values in the work are normalized, but a more quantitative measure of CssDNA would be useful addition for benchmarking purposes. IN additiong mg/L of protein produced is important to describe.

We would like to thank the reviewer for this comment. In order to address this point, we first compared the expression levels of CssDNA with its corresponding plasmid and linear double-stranded expression templates. We obtained the linear double-stranded expression templates by PCR (1210bp, which encompass only the gene cassette, named PCR) and single endonuclease digestion of the plasmids (4358bp, named L-pl). As shown in Figure 1a, different double-stranded expression templates and CssDNA were characterized by agarose gel electrophoresis. We added the same mass concentration of CssDNA and double-stranded templates to a certain reaction system, respectively, and

observed the change of fluorescence signal with time and the final protein expression level. From the fluorescence protein expression kinetic curves of C_{ss}DNA and its corresponding L-pl and PCR fragments, it can be found that the expression levels of traditional double-stranded DNA expression templates are much better than C_{ss}DNA (Figure 1b-e). The higher expression level of PCR is the result of more moles at the same mass concentration. Moreover, the expression rate of the double-stranded expression templates is faster than that of C_{ss}DNA (Figure 1b, 1d). There are two main reasons for the difference in expression between C_{ss}DNA templates and traditional double-stranded templates. The first reason is that C_{ss}DNA expressing protein must first convert the single-stranded form to the double-stranded form (or partially double-stranded form) for subsequent transcription and translation, while the double-stranded templates can be directly transcribed and translated. The replication process of C_{ss}DNA templates is the key limiting step of its expression rate and expression level. The second reason is that commercially available protein expression kits have only been optimized and developed for transcription and translation processes¹. Therefore, existing reaction systems is not conducive to the expression of proteins using C_{ss}DNA as the expression template.

Figure 1. Comparison of expression level of traditional templates and C_{ss}DNA template.

a. 1% agarose gel analysis of different expression templates. **b, c.** Expression level of CssDNA(+) and its corresponding double-stranded DNA. **d, e.** Expression level of CssDNA(-) and its corresponding double-stranded DNA. The final concentration of different templates is 1ng/uL. Error bars represent standard deviation of at least three independent tests.

To further demonstrate the comparison of the technique with the existing state-of-the-art technique in the same experiment, we added a more detailed supplement to Fig S4. The effect of aphidicolin (a tetracyclic diterpenoid) on protein expressed by different templates is shown in Figure 2. All fluorescence signals were normalized according to the fluorescence intensity of the highest expression level of the corresponding expression template. The results show that aphidicolin have a significant inhibitory effect on CssDNA expression (Figure 2a-b, 2d-e), but do not inhibit protein expression from plasmid (Figure 2a, 2c, 2d, 2f), which is one of the differences between CssDNA and traditional templates. In other words, the expression level of CssDNA template is easier to regulate than the plasmid template.

Figure 2 (Corresponding to Fig. S6 in the revised Supplementary materials). Effect of aphidicolin on protein expression of the CssDNA and plasmid vectors. **a, d.** Effect of aphidicolin on expression level of CssDNA(+) and plasmid(+) (**a**), CssDNA(-) and plasmid(-) (**d**). **b, c.** Changes in protein expression levels over time for CssDNA(+) (**b**) and plasmid(+) (**c**) vectors in the presence of aphidicolin. **e, f.** Changes in protein expression levels over time for CssDNA(-) (**e**) and plasmid(-) (**f**) vectors in the presence of aphidicolin. The final concentration of different templates is 1ng/uL. All fluorescence signals were normalized according to the fluorescence intensity of the highest expression level of the corresponding expression template. Error bars represent standard deviations from at least three independent tests. Statistical analysis was performed using two-way ANOVA with Tukey's

multiple comparison (* $p \leq 0.05$, ** $p \leq 0.01$, *** $p \leq 0.001$, **** $p \leq 0.0001$, ns $p > 0.05$).

To provide a more quantitative CssDNA expression results, CssDNA protein expression kinetic curves before normalization were added to the revised supplementary material (Figure S2-S3), as shown in Figure 3.

In addition, the protein produced was purified and quantified as the reviewer suggested. The expressed protein was His-tagged, so we used His-Monster Beads (purchased from Kangma-Healthcode, Shanghai) to purify the produced protein. When the protein expression level of CssDNA template reached a plateau (that is, the expression stopped), 1mL of reaction mixture was taken. After centrifugation, the supernatant containing EGFP was added to the magnetic beads. The beads bind with target protein were washed three times, and protein was eluted. The gels of coomassie blue staining and fluorescence imaging showed that the expression EGFP was purified (Figure 4). The protein was quantified by bicinchoninic acid (BCA) assay after overnight dialysis. The result showed that when the final concentration of CssDNA(+) was 1ng/uL, the protein yield was 10mg/L, and the protein mass of plasmid(+) was 40mg/L at the same concentration. It's worth noting that this yield is not the highest yield, and the protein yield is related to template concentration and reaction volume.

Figure 3 (Corresponding to Fig. S2, S5 in the revised Supplementary materials). **a, b.** Changes in fluorescence intensity over time for CssDNA(+) (**a**) and CssDNA(-) (**b**) vectors (5ng/uL) during expression relative to the blank group. **c, d.** Changes in fluorescence intensity over time for CssDNA(+) (**c**) and CssDNA(-) (**d**) vectors (5ng/uL) in the presence of dNTPs or aphidicolin. Error bars represent standard deviation of at least three independent tests.

Figure 4 (Corresponding to Fig. S3 in the revised supplementary materials). Characterization of protein purification process. NC: blank reaction mixture as negative control. Rxn: reaction mixture containing proteins expressed by C_{ss}DNA template. Supernatant: supernatant obtained by Rxn centrifugation. FT: liquid flowing through magnetic beads. Wash 1-3: liquid after washing magnetic beads 1-3 times. Elution (denatured): the proteins in elution buffer was denatured by heating. Elution (nondenatured): the proteins in elution buffer wasn't denatured.

- If similar, I just am not sure if people would switch to using this approach, which lessens my enthusiasm for having the work published in a high impact journal.

Regarding the reviewer's concerns on the uniqueness of C_{ss}DNA expression templates, we appreciate that this point must be emphasized in this work. Here, we would like to further highlight three impactful aspects of C_{ss}DNA that are significantly different from traditional templates. The first uniqueness of C_{ss}DNA is the fate of expression. Unlike plasmids, C_{ss}DNA must convert its single-stranded form into a double-stranded form (or partially double-stranded form). The required replication process provides more opportunities to regulate the expression level of C_{ss}DNA. For example, the addition of aphidicolin, a DNA polymerase inhibitor, can inhibit the expression of C_{ss}DNA, but not the plasmid (Figure 5). In addition, all factors associated with the replication process have the potential to regulate the expression of C_{ss}DNA template.

Secondly, the expression of anti-sense C_{ss}DNA can be regulated by T7 complementary strand. As shown in Figure 6, the addition of T7 complementary strand can promote the expression of C_{ss}DNA(-), because it directly initiates transcription process. Therefore, we can regulate the expression of C_{ss}DNA(-) by controlling the expression pathway of C_{ss}DNA(-). Based on this feature, a variety of logic gates can also be constructed through designing different T7 complementary strands (Figure 4 of the manuscript). However, plasmids exist in a double-stranded form and can be directly transcribed, so their expression cannot be regulated in this way.

The third uniqueness of C_{ss}DNA is the addressability and programmability. C_{ss}DNA can be used as the scaffold DNA to form DNA nanostructures of various geometries, such as

DNA origami²⁻³. Take CssDNA(-) for example, CssDNA(-) can be folded into six helix bundles along predetermined paths with the aid of hundreds of short “staple” strands. The DNA origami formed by CssDNA(-) was characterized by agarose gel electrophoresis and transmission scanning electron microscope(TEM) (Figure 7a-b). To compare the expression level of CssDNA(-) and DNA origami formed by CssDNA(-), the DNA origami was ultrafiltration to remove excess staple strands, and an equal amount of staples were added to CssDNA(-). We found that the expression of CssDNA(-) folded into DNA origami was significantly inhibited, due to the complex nanostructure interfering with the replication process (Figure 7c-e). Obviously, plasmids cannot form into DNA nanostructures, nor can their expression level be regulated in this way. (Please note that the experimental results of DNA origami affecting CssDNA expression are to be published later.)

Although the current expression level of CssDNA is not as good as plasmids, as a new toolbox of synthetic biology, it provides more ways to regulate protein expression and promotes the development of synthetic biology.

Figure 5 (Same as Fig. 2) Effect of aphidicolin on protein expression of the CssDNA and plasmid vectors. **a, d.** Effect of aphidicolin on expression level of CssDNA(+) and plasmid(+) (a), CssDNA(-) and plasmid(-) (d). **b, c.** Changes in protein expression levels over time for CssDNA(+) (b) and plasmid(+) (c) vectors in the presence of aphidicolin. **e, f.** Changes in protein expression levels over time for CssDNA(-) (e) and plasmid(-) (f) vectors in the presence of aphidicolin. The final concentration of different templates is 1ng/uL. All fluorescence signals were normalized according to the fluorescence intensity of the highest expression level of the corresponding expression template. Error bars represent standard deviations from at least three independent tests. Statistical analysis was performed using two-way ANOVA with Tukey's multiple comparison (* $p \leq 0.05$, ** $p \leq 0.01$, *** $p \leq 0.001$, **** $p \leq 0.0001$, ns $p > 0.05$).

Figure 6 (Corresponding to Fig. S10 in the revised supplementary materials). Effect of T7 complementary strand on protein expression of the CssDNA(-) and plasmid(-). **a**. Comparison of expression level of CssDNA(-) and plasmid(-) in the presence of T7 complementary strand. **b, c**. Changes in protein expression levels over time for CssDNA(-) (5ng/uL) (**b**) and plasmid(-) (1ng/uL) (**c**) vectors in the presence of aphidicolin. All fluorescence signals were normalized according to the fluorescence intensity of the highest expression level of the corresponding expression template. Error bars represent standard deviations from at least three independent tests. Statistical analysis was performed using two-way ANOVA with Tukey's multiple comparison (* $p \leq 0.05$, ** $p \leq 0.01$, *** $p \leq 0.001$, **** $p \leq 0.0001$, ns $p > 0.05$).

Figure 7. Effect of DNA origami on protein expression of the CssDNA(-) and plasmid(-). **a, b**. Agarose gel electrophoresis (**a**) and TEM (**b**) were used to characterize the folding of CssDNA(-) into DNA origami. **c**. Comparison of expression level of CssDNA(-) folded into DNA origami and plasmid(-). **d, e**. Changes in protein expression levels over time for CssDNA(-) folded into DNA origami (2ng/uL) (**d**) and plasmid(-) (1ng/uL) (**e**). All fluorescence signals were normalized according to the fluorescence intensity of the highest expression level of the corresponding expression template. Error bars represent standard

deviations from at least three independent tests. Statistical analysis was performed using two-way ANOVA with Tukey's multiple comparison (* $p \leq 0.05$, ** $p \leq 0.01$, *** $p \leq 0.001$, **** $p \leq 0.0001$, ns $p > 0.05$).

- The authors claim that C_{ss}DNA (+) is not able to be transcribed directly, but rather must be converted to dsDNA by DNA polymerase first, and then transcribed. It would be useful to prove this directly by in vitro transcription reaction, as was done for C_{ss}DNA (-). Also, if this is true, why not just use double stranded DNA?

We appreciate the reviewer's comment on the need to supplement the in vitro transcription of C_{ss}DNA(+). As shown in Figure 8, C_{ss}DNA(+) itself cannot be directly transcribed. The results of in vitro transcription further demonstrated that C_{ss}DNA(+) must be converted to dsDNA by DNA polymerase and then transcribed. Although double-stranded DNA (such as plasmids and PCR fragments) can be directly transcribed, this limits how dsDNA expression can be regulated. In contrast, the expression regulation of single-stranded DNA is more diverse than double-stranded DNA, as we discussed above (Figure 5-7).

Figure 8 (Corresponding to Fig. 2 in the revised manuscript). 1.5% agarose gel analysis of mRNA obtained by in vitro transcription of C_{ss}DNA(-) that combined with different T7 complementary strands, as well as C_{ss}DNA(+).

- The discussion around the mechanism proposed in Fig 3g with incomplete DNA replication intermediates is not well explained. The term "DNA replication intermediates" is suddenly introduced and is poorly defined. It seems that the authors are saying some of these replication intermediates will look like T7 complementary strands fused to C_{ss}DNA through chance, and that subset can be directly transcribed? The authors should explain this section more thoroughly and give more explicit reasonings behind their interpretation of the data in Fig 3.

We apologize for any confusion caused by this description in our initial manuscript and fully agree with the reviewer to give a clearer explanation. We also added a more detailed description to explain the results and support our conclusions. The revised parts of the manuscript are indicated in red.

We have changed the wording here to explain “DNA replication intermediates” as follows: “...we conclude that not all expression of C_{ss}DNA(-) requires a complete replication process and that a transcription process is present. We thus surmise that, C_{ss}DNA(-) expression in the presence of aphidicolin is primarily due to direct transcription from the C_{ss}DNA starting from incomplete DNA replication intermediates (i.e., partially hybridized intermediates of DNA replication, in which the T7 promoter is present in double-stranded form), which is a gene expression pathway that differ from that for C_{ss}DNA(+)....”

Minor Criticisms:

- In lines 49-42 the authors claim that it is “infeasible” to create complex genetic circuits from double stranded DNA and cite a paper on mammalian circuits and allosteric transcription factors. This is a large claim to make with minimal support. Please reword or expand with better citations.

Thanks very much for the reviewer’s suggestions. We have changed the wording here and cited more literature to support our claim. The changes are as follows:

“Although switchable molecular devices have been achieved through tuning the recognition and binding of transcription factors on the double-stranded DNA, the limited number of available regulatory factors translates to a relatively low number of methods for regulating double-stranded DNA expression. This makes it difficult to construct more complex genetic circuits based on double-stranded DNA vectors alone⁴⁻⁹”

The citation here shows that allosteric transcription factors, which are characterized extensively in cells, are ported over to CFE (cell-free expression) systems. And they are widely used as biosensors to regulate protein expression. It also shows that transcription factors are the most commonly used regulation methods for double-stranded DNA.

- Do the authors have hypotheses on why C_{ss}DNA (+) and (-) behave so differently? Do the authors have an idea for the mechanistic reason that C_{ss}DNA (+) is unable to be transcribed directly (or from intermediates/T7 complementary strand fusion) and must only rely on replication? Why are the expression pathways different?

The differences between C_{ss}DNA(+) and C_{ss}DNA(-) are determined by the sequence. C_{ss}DNA(+) (sense C_{ss}DNA strand), also known as the coding strand or the plus strand, determines the protein-coding sequence. The sequence of C_{ss}DNA(+) is similar to the resulting mRNA, except for the substitution of uracil(U) for thymine(T). C_{ss}DNA(-) (antisense C_{ss}DNA strand), also referred to as the template strand or the minus strand, plays an important role in RNA synthesis. It acts as the template for RNA synthesis, guiding the formation of mRNA. During transcription, RNA polymerase reads C_{ss}DNA(-) from 3’ to 5’, while RNA polymerase does not read C_{ss}DNA(+). Therefore, C_{ss}DNA(+) must rely on replication to synthesize the template strands and then transcription.

In addition, T7 RNA polymerase is highly specific for its promoter sequence. After binding to the promoter, it is possible to make transcripts. The template for transcription can be (1) a plasmid that typically has the promoter for in vitro transcription, (2) a PCR product that

has the T7 promoter as part of the 5'-oligonucleotide, and (3) two annealed oligonucleotide that carries the T7 promoter sequence and the template to be transcribed (in this case, only the T7 promoter part of the template needs to be double-stranded)¹⁰. Therefore, C_{ss}DNA(-) can be transcribed directly when the T7 promoter is present in double-stranded form.

- There is no Y axis title for Fig S10.

Thank you very much for pointing out the mistake. We have added the Y axis title "Fluorescence (Norm.)" to Fig. S15 (original Fig. S10).

- Further explain what the normalization scheme is for fluorescence. What the "equivalent C_{ss} DNA expression plateau" is not clear.

Thanks very much for the reviewer's suggestions. When the expression reached the plateau, the protein expression stopped and the fluorescence intensity was the highest. Therefore, the fluorescence signals were normalized according to the fluorescence intensity of the highest expression level of the corresponding expression template.

We changed the wording to "...all fluorescence signals were normalized according to the fluorescence intensity of the highest expression level of the corresponding expression vector...".

- In line 177 the authors state there are 13 DNA strands designed, but list 14 designs P1-P14. Is this a typo or is one a control?

We apologize for any confusion caused by this description in our initial manuscript. The 13 DNA strands we designed around the T7 promoter region but not include the T7 promoter. The DNA strand that complement to T7 promoter region was denoted by P5.

We have changed the wording to "...we designed another thirteen 19-nt DNA strands (denoted by P1, P2, P3...P14, respectively) around T7 promoter region to hybridized with C_{ss}DNA(-) (Fig. 2h). Among them, P1-P4 were located in front of T7 promoter region, and P6-P14 were located behind T7 promoter....".

Reviewer #3 (Remarks to the Author):

Tian et al present a novel component for cell-free expression systems in their manuscript in the form of circular single stranded DNA (C_{ss}DNA) templates. They study the protein expression kinetics of C_{ss}DNA templates carrying sense and antisense genes and identify the dominant pathways for both and different components, such as complementary nucleotides and DNA synthesis inhibitors that they can use to exploit these to create logic gates, demonstrating C_{ss}DNA's applicability in reaction circuits. In my opinion the C_{ss}DNA templates presented in this work can be a useful, new addition in the toolbox for generating gene circuits that have clear advantages compared to double stranded templates as they can be more easily be integrated into oligonucleotide reaction circuits therefor, I find the works contribution to the field of synthetic biology significant. I have found the experiments performed to be quite thorough and convincing and to support the conclusions drawn upon them. However, I have a number of comments and points the authors should address before I would recommend the manuscript to be accepted for publication.

We thank reviewer #3 for their constructive remarks, which enabled us to significantly improve our revised manuscript. According to the reviewer's comments, we have made the changes and further improved the manuscript, as detailed below.

1. The authors use phage derived ssDNA as their template. While it is quite an efficient way to produce ssDNA, in my opinion it potentially introduces some unneeded complexity. There are other ways to generate long ssDNA molecules that would only or mostly contain the gene-cassette e.g., with RCA (Nat Methods 10, 647–652 (2013)), asymmetric (Sci Rep 9, 6121 (2019)) or other PCR approaches (Nano Lett. 2009, 9, 12, 4302–4305) or by using a much smaller scaffold to clone in (Nanoscale, 2015,7, 16621-16624). If the template would only or mostly contain the gene some of the observed concentration dependent negative effects would be reduced possibly as more products would be complete. While the comparison with templates generated with the RCA and mini scaffold approach is out of the scope of this work, I think it would be relatively easy to produce a single stranded templates encompassing the gene cassette using one of the PCR approaches and compare its behavior to the C_{ss}DNA, therefor I think the authors should perform these experiment in addition to comparing their approach to these alternative approaches in the text.

Thanks very much for the reviewer's comments. Here we produced ssDNA using streptavidin-coated magnetic beads¹¹. The desired fragment (which encompasses only the gene cassette, 1210bp) was amplified by PCR using a biotinylated primer to generate the complement of the desired strand of DNA and a non—biotinylated primer for a strand of interest. Then the PCR products were bound to streptavidin magnetic beads and isolated using a magnetic bead separator. After exposure to a melting solution that denatures the dsDNA, the undesired (biotinylated) strand of DNA was removed by another round of magnetic separation. After the neutralization of the supernatant, the L_{ss}DNA (linear single-

stranded DNA) was pelleted and resuspended. To ensure the purity of LssDNA, the products were purified again by gel electrophoresis.

As shown in Figure 9a, pure LssDNA(+) and LssDNA(-) were obtained. According to the expression level of CssDNA(+) and CssDNA(-) at different concentrations, we chose a lower concentration (1ng/uL) to compare sense strands and a higher concentration (10ng/uL) to compare antisense strands. Figure 9b and c demonstrated that the expression level of LssDNA is much lower than that of CssDNA. The difference in expression between LssDNA and CssDNA is mainly due to their different stability. Systematic studies have shown that circular DNA has higher nuclease resistance and higher stability than linear single-stranded DNA¹²⁻¹⁴. Moreover, DNA nucleases native to cells are present in the crude cell lysate and are retained after the lysate is purified¹⁵. These nucleases readily digest linear single-stranded DNA fragments in the reaction, resulting in a much shorter half-life for LssDNA than for CssDNA, as well as greatly reduced protein production. This is also one of the reasons why we chose CssDNA instead of LssDNA.

Figure 9. Comparison of LssDNA and CssDNA. **a.** 1% agarose gel analysis of dsDNA, LssDNA and CssDNA. **b.** The expression level of LssDNA(+) and CssDNA(+) at the concentration of 1ng/uL. **c.** The expression level of LssDNA(-) and CssDNA(-) at the concentration of 10ng/uL. Error bars represent standard deviation of at least three independent tests.

2. The authors perform a number of experiments (e.g. in Fig.1 e/f/h/l, Fig.2 b/c/e) where they follow the gene expression in time and report on the differences in the observed kinetics. The authors should fit these curves to derive apparent kinetic parameters (e.g. rate constants) and report those in the text.

We would like to thank the reviewer for this comment. We obtained the maximum rate constant (k_{max}) by taking the first derivative of the fluorescence kinetic curve. The maximum rate constants corresponding to the fluorescence curves in manuscript are shown in Figure 10 and Figure 11. And it is a good supplement to the conclusion of this paper.

Figure 10 (Corresponding Fig.S3 in the revised supplementary material). **a, c.** Maximum rate constants of the fluorescence kinetic curve of CcssDNA(+) (**a**) and CcssDNA(-) (**c**). **b, d.** Maximum rate constants of the fluorescence kinetic curve of CcssDNA(+) (**b**) and CcssDNA(-) (**d**) in the presence of aphidicolin or dNTPs.

Figure 11 (Corresponding Fig.S9 in the revised supplementary material). **a, b.** Maximum rate constants of the fluorescence kinetic curve of CcssDNA(+) (**a**) and CcssDNA(-) (**b**) in the presence of T7 complementary strand. **c.** Maximum rate constants of the fluorescence kinetic curve of CcssDNA(-) bound to different T7 complementary strands.

3.The results presented in Fig. 2e/f/g are somewhat perplexing. Based on the results presented in Fig2.f the 9nt and 13nt T7 oligos produce a much lower amount of RNA than the rest of the oligos. The other should by the way quantitate this and report it in a plot. Taken this difference in transcription yield it is quite remarkable the expression of the

protein seems to have similar yield (based on the saturation value) and rate (based on the slope of the linear phase of the curve). The authors should comment on this.

We fully agree with this reviewer's suggestion. We supplemented the in vitro transcription results of C_{ss}DNA(+) (Figure 12, lane 8). To further quantify the results of in vitro transcription, we quantified the gray values of the gel electrophoresis bands using Image J and provided the mRNA concentrations determined by nanodrop, as shown in Figure 12b and c. We can find that the transcription capacity of C_{ss}DNA(-)+T7 (C_{ss}DNA(-) bound to T7 complementary strand) decreased with the decrease of the T7 complementary strand length.

In addition, we designed a novel C_{ss}DNA whose transcript contains two spinach aptamers following the EGFP sequence (C_{ss}-2F30, Figure 13a). The fluorogenic aptamer Spinach is a short RNA sequence that can bind to DFHBI ((Z)-4-(3,5-difluoro-4-hydroxybenzylidene)-1,2-dimethyl-1H-imidazol-5(4H)-one), a small-molecule mimic of the intrinsic chromophore of GFP. In isolation, DFHBI is not fluorescent, but it exhibits fluorescence comparable to that of GFP upon binding to the RNA¹⁶⁻¹⁸. Therefore, we further quantified in vitro transcribed RNA by fluorescence of DFHBI binding RNA. As shown in Figure 13d, the fluorescence intensity demonstrated changes of transcription capacity in the presence of different T7 complementary strands. This is consistent with the result of gel electrophoresis (Figure 13c).

As shown in Figure 13b, T7 complementary strands with different length have the similar promoting effects, which is consistent with the results in manuscript. The similar promoting effect is due to the presence of excess templates in reaction system, that is, the amount of RNA transcribed by C_{ss}DNA(-)+T7 (C_{ss}DNA(-) combined with T7 complementary strand) is much greater than the RNA required for translation. Therefore, protein expression levels cannot indicate the transcription capacity of C_{ss}DNA(-)+T7.

Figure 12 (Corresponding to Fig. 2 in the revised manuscript and Fig.S12 in the revised supplementary material). Quantification of mRNA transcribed by C_{ss}DNA(-) in vitro. **a**. 1.5% agarose gel analysis of mRNA obtained by in vitro transcription of C_{ss}DNA(-) that combined with different T7 complementary strands, as well as C_{ss}DNA(+). **b**. Gray values of the gel electrophoresis bands were quantified using Image J. **c**. The mRNA concentrations measured by nanodrop.

Figure 13 Expression levels and in vitro transcription levels of Css-2F30. **a.** The production of Css-2F30 was characterized by agarose gel electrophoresis. Lanes 1-6 are plamid-2F30(+), plamid-2F30(-), Css-2F30(+), Css-2F30(-), CssDNA(+), and CssDNA(-), respectively. **b.** The changes in Css-2F30(-) protein expression over time after the addition of T7 complementary strands of different lengths. **c.** 1.5% agarose gel analysis of mRNA obtained by in vitro transcription of Css-2F30(-) that combined with different T7 complementary strands, as well as Css-2F30 (+). **d.** Fluorescence of DFHBI combined with transcribed mRNA. Error bars represent standard deviation of at least three independent tests.

4. The results presented in Fig. 4 are a nice demonstration of the system's capability to create logic gates. However, some fundamental logic gates, such as AND gates, are not demonstrated, which could be important to implement to show general applicability, using perhaps input oligos with partially complementary sequences to stabilize their binding to the promoter through three way junctions. Additionally, the authors claim that they create gene circuits using CssDNA when they create only logic gates in the strict sense. The authors should change this in the text.

We agree with the reviewer that we should have been more precise in our wording, using "logic gates" instead of "logic circuits". We appreciate the reviewer's comment on the need to further show general applicability of our system. Here, we constructed two fundamental logic gates, NAND gate and AND gate.

To construct the NAND gate, we obtained the initial gate structure by annealing the CssDNA(-) with a longer T7 complementary strand that had toehold sequence at the 5' end. Each input has two parts, the first part is partially complementary to T7 complementary strand, and the second part is partially complementary to the other input. None of them can displace the T7 complementary strands from the initial gate structure. Fluorescence

was high either in the absence of any input or in the presence of only one input. When both inputs are present, they can form three-way structure with T7 complementary strand, releasing C_{ss}DNA(-) from the initial gate structure (Figure 14a).

For the AND gate, we tried to design the logic gate based on the reviewer's comments. We have shown that 9nt T7 complementary strand can effectively promote protein expression and its location is in the middle of T7 promoter (Figure 2 in manuscript). Therefore, the design of the AND gate input needs to consider the length and position of DNA strand. Due to these restrictions, the several attempt we made were not very good. So we constructed the AND gate in a different way. A specially designed double-stranded DNA was introduced into the initial gate. One of the strands is a T7 complementary strand, the other is a T7 blocking strand with the same sequence as C_{ss}DNA(-). The input was designed to complement the T7 blocking strand. When there are no inputs or one of the inputs exists, T7 complementary strand was blocked and the fluorescence is low. In the presence of both inputs, T7 complementary strand can be released and the fluorescence is elevated (Figure 14b). The higher background is due to the input sequence being the same as T7 complementary strand.

Figure 14 (Corresponding to Fig.S26 in the revised supplementary material). Construction of logic gates using C_{ss}DNA as a logic element. **a, b**. Schematic and fluorescence signals of two-input logic gates (including NAND and AND) under different input combinations. Error bars represent standard deviation of at least three independent tests.

1. Silverman, A. D.; Karim, A. S.; Jewett, M. C. Cell-free gene expression: an expanded repertoire of applications. *Nat. Rev. Genet.* 2019, 21 (3), 151-170.
2. Dey, S.; Fan, C.; Gothelf, K. V.; Li, J.; Lin, C.; Liu, L.; Liu, N.; Nijenhuis, M. A. D.; Saccà, B.; Simmel, F. C.; Yan, H.; Zhan, P. DNA origami. *Nat. Rev. Methods Primers* 2021, 1 (1), 13.
3. Zhang, Y.; Pan, V.; Li, X.; Yang, X.; Li, H.; Wang, P.; Ke, Y. Dynamic DNA Structures. *Small* 2019, 15 (26), e1900228.
4. Li, S.; Li, Z.; Tan, G.-Y.; Xin, Z.; Wang, W. In vitro allosteric transcription factor-based biosensing. *Trends Biotechnol.* 2023, 41 (8), 1080-1095.
5. Lienert, F.; Lohmueller, J. J.; Garg, A.; Silver, P. A. Synthetic biology in mammalian cells: next generation research tools and therapeutics. *Nat. Rev. Mol. Cell. Biol.* 2014, 15 (2), 95-107.
6. Liu, X.; Silverman, A. D.; Alam, K. K.; Iverson, E.; Lucks, J. B.; Jewett, M. C.; Raman, S. Design of a transcriptional biosensor for the portable, ondemand detection of cyanuric acid. *ACS Synth. Biol.* 2020, 9(1), 84–94
7. Wen, K. Y.; Cameron, L.; Chappell, J.; Jensen, K.; Bell, D. J.; Kelwick, R.; Kopniczky, M.; Davies, J. C.; Filloux, A.; Freemont, P. S. A Cell-Free Biosensor for Detecting Quorum Sensing Molecules in *P. aeruginosa*-Infected Respiratory Samples. *ACS Synth. Biol.* 2017, 6 (12), 2293-2301.
8. Karig, D. K.; Iyer, S.; Simpson, M. L.; Doktycz, M. J., Expression optimization and synthetic gene networks in cell-free systems. *Nucleic Acids Res.* 2012, 40 (8), 3763-3774.
9. Kawaguchi, T.; Chen, Y. P.; Norman, R. S.; Decho, A. W. Rapid Screening of Quorum-Sensing Signal N-Acyl Homoserine Lactones by an In Vitro Cell-Free Assay. *Appl. Environ. Microb.* 2008, 74 (12), 3667-3671.
10. Beckert B.; Masquida B. Synthesis of RNA by in vitro transcription. *Methods Mol. Biol.* 2011, 703, 29-41.
11. Wakimoto Y, Jiang J, Wakimoto H. Isolation of single-stranded DNA. *Curr. Protoc. Mol. Biol.* 2014 ,1(107), 2.15.1-2.15.9.
12. Kuai, H.; Zhao, Z.; Mo, L.; Liu, H.; Hu, X.; Fu, T.; Zhang, X.; Tan, W. Circular Bivalent Aptamers Enable in Vivo Stability and Recognition. *J. Am. Chem. Soc.* 2017, 139 (27), 9128-9131.
13. Li, J.; Zhou, J.; Liu, T.; Chen, S.; Li, J.; Yang, H. Circular DNA: a stable probe for highly efficient mRNA imaging and gene therapy in living cells. *Chem. Commun.* 2018, 54 (8), 896-899.
14. Pan, X.; Yang, Y.; Li, L.; Li, X.; Li, Q.; Cui, C.; Wang, B.; Kuai, H.; Jiang, J.; Tan, W. A bispecific circular aptamer tethering a built-in universal molecular tag for functional protein delivery. *Chem. Sci.* 2020, 11 (35), 9648-9654.
15. McSweeney, M. A.; Styczynski, M. P. Effective Use of Linear DNA in Cell-Free Expression Systems. *Front. Bioeng. Biotechnol.* 2021, 20(9), 715328.
16. Wu, J.; Zaccara, S.; Khuperkar, D.; Kim, H.; Tanenbaum, M. E.; Jaffrey, S. R. Live imaging of mRNA using RNA-stabilized fluorogenic proteins. *Nat. Methods* 2019, 16 (9), 862-865.
17. Jeremy S. Paige,; K. Y. W.; Samie R. Jaffrey. RNA Mimics of Green Fluorescent Protein. *Science.* 2011,333(6042), 642-646.

18. Warner, K. D.; Chen, M. C.; Song, W.; Strack, R. L.; Thorn, A.; Jaffrey, S. R.; Ferré-D'Amaré, A. R. Structural basis for activity of highly efficient RNA mimics of green fluorescent protein. *Nature Structural & Molecular Biology* 2014, 21 (8), 658-663.

REVIEWER COMMENTS

Reviewer #1 (Remarks to the Author):

I would like to thank the authors for their efforts and appreciate the improvements to the paper. I still like the concept of circular single stranded DNA (C_{ss}DNA) templates for cell-free systems.

In the rebuttal, if I understand correctly, the authors in figure 1 add the same mass of linear DNA and the circular, linearized template. This should be done in moles as it might show their system in a better light. The authors comment that the use of mass might be one reason that the PCR templates are better, but if the PCR templates are better, they are easier to use and that is what researchers in the field will use.

In figures 1c and 1e, it is not OK to show a broken line to make the authors' approach seem closer to standard PCR templates than it is. The Y-axis needs to be continuous so one can see the order of magnitude difference.

If the cell-free system is not conducive to circular templates as discussed in response to figure 1, then I believe it is a responsibility of the authors to optimize the system to show their approach is at least as good, or better than the state of the art. Then, the benefits of the C_{ss}DNA that they described would be real.

Given these features of the system, while I would love to support the work and the authors have done a lot of work, my enthusiasm is lowered.

Reviewer #3 (Remarks to the Author):

The authors have performed all the requested experiments and applied the necessary changes in the manuscript's text therefore I would recommend the publication of the revised manuscript.

Response to Reviewers-Revision 2

REVIEWER COMMENTS

Reviewer #1 (Remarks to the Author):

I would like to thank the authors for their efforts and appreciate the improvements to the paper. I still like the concept of circular single stranded DNA (C_{ss}DNA) templates for cell-free systems.

We thank you for all your previous and new suggestions which were highly valuable for us. We have implemented the corrections and optimizations as below.

In the rebuttal, if I understand correctly, the authors in figure 1 add the same mass of linear DNA and the circular, linearized template. This should be done in moles as it might show their system in a better light. The authors comment that the use of mass might be one reason that the PCR templates are better, but if the PCR templates are better, they are easier to use and that is what researchers in the field will use.

In figures 1c and 1e, it is not OK to show a broken line to make the authors' approach seem closer to standard PCR templates than it is. The Y-axis needs to be continuous so one can see the order of magnitude difference.

We fully agree with the reviewer's comment on the need to supplement the comparison of circular single-stranded DNA (C_{ss}DNA) and the circular, linear double-stranded DNA in moles. To show the expression levels of different templates in the system, we compared the expression levels of C_{ss}DNA to its corresponding plasmid and PCR product at the same mass (50ng) and the same molar amount (0.1 pmol). As shown in Figure 1, the expression level of the PCR product was higher than that of C_{ss}DNA and plasmid at the same mass, while at the same molar amount, the expression level of the plasmid was comparable to that of the PCR product. This supports our previous analysis that a higher molar amount results in a higher expression level, and it also suggests that double-stranded DNA templates (plasmid and PCR products) have similar expression capacities at the same number of moles. In this cell-free system, the expression level of C_{ss}DNA is indeed an order of magnitude worse than that of the double-stranded DNA template.

Figure 1. Comparison of expression level of traditional templates and CcssDNA template. **a, b.** Expression level of CcssDNA and its corresponding double-stranded DNA at the same mass (50ng). **c, d.** Expression level of CcssDNA and its corresponding double-stranded DNA at the same molar amount (0.1pmol). Data is presented as mean \pm s.d. with individual data overlaid representing biologically independent experiments.

If the cell-free system is not conducive to circular templates as discussed in response to figure 1, then I believe it is a responsibility of the authors to optimize the system to show their approach is at least as good, or better than the state of the art. Then, the benefits of the ccssDNA that they described would be real.

Given these features of the system, while I would love to support the work and the authors have done a lot of work, my enthusiasm is lowered.

Thank you for this important comment. As we presented in our previous manuscript, the major feature of interest of our CcssDNA is its switchability and programmability, not the expression level. Therefore, we initially did not put too much effort on the optimization of the cell free system for CcssDNA expression. In fact, as our studies showed that the CcssDNA had to be complemented to become dsDNA to be expressed (at least in the promoter region), we did not expect a higher expression level in the first place.

Nevertheless, motivated by the reviewer's comment, we now set out to optimize the cell-free system as much as possible to improve expression also from the CcssDNA template. We optimize three reaction systems based on the original commercial system by adding additional components to speed up the DNA replication, namely buffer 1, buffer 2 and buffer 3. We compare the expression levels of CcssDNA and plasmid of the same mass and the same molar amount in different reaction systems, respectively, as shown in Figure 2a-c. Although the expression level of CcssDNA has not yet reached that of the plasmid, after our optimization, the protein expression level of CcssDNA in the final reaction system (buffer 3) has been increased almost 10 times compared to the original commercial buffer,

narrowing the gap with the plasmid to some extent. The additional components added to the different reaction systems are tabulated in Figure 2d. This figure 2 has been added to the supporting information for the readers.

Figure 2 (Corresponding to Fig. S5 in the revised Supplementary materials). Optimization of the reaction system. **a**. Changes in protein expression levels over time for CssDNA and its corresponding plasmid under different reaction systems. All templates have the same mass (50ng). **b**, **c**. Comparison of fluorescence intensity of EGFP produced by CssDNA and plasmid with the same mass (50ng, **b**) and the same molar amount (0.1pmol, **c**) in different reaction systems. Data is presented as mean \pm s.d. with individual data overlaid representing biologically independent experiments. **d**. The components of different reaction systems.

The expression level of CssDNA in the current optimized cell free systems is still not better than the state of art with dsDNA, but we believe it might be further improved in the future. As a closed reaction system, the in vitro expression system only has a finite amount of substrate, its components degrade over time, and it does not allow for exchange of material and energy, which limits overall protein expression. Conversely, inside living cells, there is a continuous supply of matter and energy. Our previous work has compared the expression of CssDNA and plasmids in different mammalian cells and found that in many cells, the expression level of CssDNA can be comparable to or even exceed that of plasmids (Figure 3)¹. We therefore believe that extending the lifetime of the cell-free expression reaction, and potentially operating in an open reactor might also bring up expression levels from CssDNA further.

Back to the highlight of this work, we found that CssDNA, as a novel gene expression vector in a cell-free system, can be programmed to regulate its gene expression in a variety of ways that cannot be achieved with traditional templates. Our findings thus advance the understanding of CssDNA gene expression and expand the toolbox of gene circuits and synthetic biology.

Figure 3. The expression efficiency of various mammalian cells transfected with CcssDNA and plasmid (each 0.5 pmol), respectively, via lipofection. Error bars represent standard deviations from at least three independent tests.

Reviewer #3 (Remarks to the Author):

The authors have performed all the requested experiments and applied the necessary changes in the manuscript's text therefore I would recommend the publication of the revised manuscript.

Thank you for your insightful feedback throughout the reviewing process.

References

1. Tang, L., Tian, Z., Cheng, J. et al. Circular single-stranded DNA as switchable vector for gene expression in mammalian cells. *Nat. Commun.* **14**, 6665 (2023).